# Expectation Consistency Loss: Rethink Confidence Calibration under Covariate Shift

**Jinzong Dong**[1]   **Zhaohui Jiang**[1]   **Bo Yang**[1]

## Abstract

Confidence calibration for classification models is vital in safety-critical decision-making scenarios and has received extensive attention. General confidence calibration methods assume training and test data are independent and identically distributed ($i.i.d.$), limiting their effectiveness under covariate shifts. Previous calibration methods under covariate shift struggle with class-wise or canonical calibrations and often rely on unstable importance weighting when density ratios are large or unbounded. Given the above limitations, this paper rethinks confidence calibration under covariate shifts. First, we derive a necessary and sufficient condition for confidence calibration under covariate shifts, named *Expectation consistency condition*, which reveals covariate shifts do not necessarily lead to uncalibrated confidence and provides a weaker condition for confidence calibration than global covariate distribution alignment. Then, utilizing *Expectation consistency condition*, this paper proposes an unsupervised domain adaptation loss to calibrate confidence of the target domain, named *Expectation consistency loss* (ECL), which is compatible with canonical calibration, class-wise calibration, and top-label calibration. Third, we prove that computing ECL loss has the same sample complexity as Expected Calibration Error (ECE) and provide a theoretically grounded mini-batch trainable scheme for ECL loss. Finally, we validate the effectiveness of our method on both simulated and real-world covariate shift datasets.

---

[1]School of Automation, Central South University, Changsha, China. Correspondence to: Zhaohui Jiang <jzh0903@csu.edu.cn>.

*Proceedings of the 43$^{rd}$ International Conference on Machine Learning*, Seoul, South Korea. PMLR 306, 2026. Copyright 2026 by the author(s).

## 1. Introduction

Modern machine learning classification models, such as deep neural networks, are becoming increasingly accurate and widely applied in safety-critical fields (LeCun et al., 2015; Jiang et al., 2023). Nevertheless, decision-making systems in such fields need not only high accuracy but also the ability to recognize when they might be wrong (Munir et al., 2023). For example, in automatic disease diagnosis, if a model has low confidence in its prediction, it should defer to a medical professional (Jiang et al., 2011). Thus, classification models should provide accurate confidence estimates alongside their predictions to reflect the true likelihood of an event. Accurate confidence is more informative than mere class labels, e.g., stating "a patient has a 70% probability of having cancer" gives doctors more actionable information than just labeling the condition as "cancer". Moreover, accurate confidence facilitates classification models to better integrate with other probabilistic models, e.g., helping active learning to select more representative samples (Han et al., 2024) and improving the generalization performance of knowledge distillation (Li & Caragea, 2023). Therefore, pursuing more accurate confidence in classification models is of great importance (Gawlikowski et al., 2023).

In recent years, confidence calibration has emerged as one of the most effective methods for producing more reliable confidence estimates and has attracted considerable attention (Guo et al., 2017; Zhang et al., 2020; Kull et al., 2019; Dong et al., 2025b). However, general confidence calibration methods typically assume that the target domain (or test set) and the source domain (or calibration set) are independent and identically distributed ($i.i.d.$). When this assumption is violated due to distribution shifts, calibration performance often deteriorates significantly (Zhu et al., 2024). Covariate shift, a common type of data distribution shift, often occurs in real-world tasks like medical diagnosis across different populations or image recognition under varying lighting conditions, where the input data distribution of models changes while the underlying relationship between inputs and outputs remains consistent (Kimura & Hino, 2024). Under covariate shift, models calibrated on the source domain frequently fail to generalize to the target domain, resulting in unreliable confidence estimates (Bickel

et al., 2009). This highlights the importance of developing confidence calibration methods that remain robust under covariate shift (Hu et al., 2024).

Currently, the mainstream confidence calibration methods under covariate shift are based on importance weighting (Pampari & Ermon, 2020; Park et al., 2020; Wang et al., 2020; 2023), which adjusts the objective function by assigning weights based on the importance of instances from the source domain, thereby guiding the model to generalize to the target domain unbiasedly (Kimura & Hino, 2024). However, it is well known that importance weighting has been criticized for its instability when the density ratio is large or unbounded (Cortes et al., 2010). Hu et al. (2024) use mixup to synthesize pseudo-target data and generalize the calibration performance from the pseudo-target data to the target domain. However, the efficacy of this method hinges primarily on the degree of similarity between the pseudo-target data and the target domain data. Furthermore, existing methods primarily address the simplest prediction-based calibration (i.e., top-label calibration). To our knowledge, there remains a notable absence of class-wise and canonical calibration methods designed to handle covariate shift.

Importance weighting in confidence calibration aims to globally align covariate distributions, inspired by accuracy improvement under covariate shift. However, confidence calibration differs fundamentally from accuracy improvement: it requires not learning new knowledge, but precisely conveying uncertainty. This raises a natural but often neglected question: **Is global covariate distribution alignment necessary?** To answer this, we first derive a necessary and sufficient condition for confidence calibration under covariate shifts, termed the *Expectation consistency condition*. This condition reveals that covariate shifts do not necessarily cause miscalibration and provides a weaker requirement than global distribution alignment. Based on this condition, we propose an unsupervised domain adaptation loss, *Expectation consistency loss* (ECL), with three variants for canonical, class-wise, and top-label calibration. We prove that ECL has sample complexity $\mathcal{O}(B/\varepsilon^2)$, comparable to histogram binning, where $B$ denotes the number of confidence bins. To enable unbiased gradient backpropagation on mini-batch data, we also provide a theoretically sound mini-batch training scheme for ECL. Finally, we validate the method on simulated and real-world covariate shift datasets.

## 2. Background and Related Work

Consider a $K$-class classification problem where $X \in \mathcal{X}$ denotes the input feature and $Y = (Y_1, \cdots, Y_K) \in \mathcal{Y}$ denotes the $K$-class one-hot encoded label variable, with $\mathcal{X} \subset \mathbb{R}^d$ and $\mathcal{Y} = \{e_k\}_{k=1}^K$, where $e_k$ is a unit vector whose $k$-th component is 1. Let $f : \mathcal{X} \to \mathcal{S} \subset \Delta_{K-1}$ be a probabilistic classifier, where $\Delta_{K-1}$ represents a $(K-1)$-

dimensional simplex. The predicted confidence score vector is given by $S = f(X) = (f_1(X), \cdots, f_K(X)) = (S_1, \ldots, S_K) \in \mathcal{S}$. In general, the true class scalar is $Y^* = \operatorname{argmax}_k \{Y_k\}_{1 \le k \le K}$, the predicted class is defined as $\hat{Y} = \operatorname{argmax}_k \{S_k\}_{1 \le k \le K}$, and the confidence score of the predicted class is $\hat{S} = \max \{S_k\}_{1 \le k \le K}$.

In covariate shift, let $P_s(\cdot)$ and $P_t(\cdot)$ denote the probability density (for continuous variables, e.g., $X$, $X|S$, $S|X$, and $X|Y$) or probability measure (for discrete variables, e.g., $Y$, $Y|S$ and $Y|X$) on the source domain and target domain, respectively. $P$ denotes either $P_s$ or $P_t$ in cases where distinguishing between the source and target domains is not required. Let $D_s$ and $D_t$ represent the source domain and target domain data, respectively.

### 2.1. Confidence Calibration

Confidence calibration aims to match the predicted confidence vector with the true posterior probability of event occurrence. Formally, we state:

**Definition 2.1. (Perfect Calibration)** A classifier is perfectly calibrated if the following equation holds:

$$P(Y_k = 1|S = s) = s_k, \forall 1 \le k \le K, \qquad (1)$$

where $s = (s_1, \cdots, s_K)$ is the observed confidence score vector on $S$.

**Remark:** Definition 2.1 considers the most stringent calibration paradigm, named canonical calibration (Dong et al., 2025a). Appendix A provides two other common calibration paradigms: top-label calibration (Guo et al., 2017) and class-wise calibration (Kull et al., 2019).

Existing general work primarily falls into two groups: train-time calibration (Liu et al., 2023; Müller et al., 2019; Fernando & Tsokos, 2022; Hebbalaguppe et al., 2022; Grathwohl et al., 2020; Yang & Ji, 2021) and post-hoc calibration (Guo et al., 2017; Kull et al., 2019; Zhang et al., 2020; Rahimi et al., 2020; Gupta et al., 2021; Dong et al., 2025b). Train-time calibration typically carries out calibration during the classifier's training by adjusting the objective function, and post-hoc calibration learns a transformation (referred to as a calibration map) of the classifier's output on a calibration dataset in a post-hoc manner. However, these methods' effectiveness hinges on the $i.i.d.$ assumption between the target and source domains. When covariate shift occurs, this i.i.d. assumption is violated, making it difficult for the methods above to effectively calibrate confidence.

### 2.2. Confidence Calibration under Covariate Shift

In covariate shift, the target domain and the source domain have different feature distributions but the same conditional distributions. Formally, we state:

*Table 1.* Comparison of ECL and related calibration methods.

| Method | Covariate Shift | Class-wise Calibration | Canonical Calibration | Density Ratio Unbounded | Mini-batch Trainable |
|---|---|---|---|---|---|
| $SB\text{-}ECE$ (Karandikar et al., 2021) | ✗ | ✗ | ✗ | ✓ | ✗ |
| $DECE$ (Bohdal et al., 2023) | ✗ | ✗ | ✗ | ✓ | ✗ |
| $ECE^{KDE}$ (Popordanoska et al., 2022) | ✗ | ✓ | ✓ | ✓ | ✓ |
| $Weighted\ TS$ (Pampari & Ermon, 2020) | ✓ | ✗ | ✗ | ✗ | ✗ |
| $FL+IW+Temp$ (Park et al., 2020) | ✓ | ✗ | ✗ | ✗ | ✗ |
| $TransCal$ (Wang et al., 2020) | ✓ | ✗ | ✗ | ✗ | ✗ |
| $DRL$ (Wang et al., 2023) | ✓ | ✗ | ✗ | ✗ | ✗ |
| $PseudoCal$ (Hu et al., 2024) | ✓ | ✗ | ✗ | ✓ | ✗ |
| $ECL$ (Ours) | ✓ | ✓ | ✓ | ✓ | ✓ |

**Definition 2.2. (Covariate Shift)** Covariate shift occurs if the following two conditions are satisfied: $P_s(X) \neq P_t(X)$ and $P_s(Y|X) = P_t(Y|X)$.

Table 1 summarizes the characteristics of related calibration methods in five key dimensions, including whether they can handle covariate shifts, whether they support class-wise/canonical calibration, whether they can handle unbounded density ratios, and whether they are theoretically mini-batch trainable. As shown, existing methods often cover only a portion of the capabilities. In contrast, our ECL satisfies all dimensions simultaneously, demonstrating the method's comprehensiveness and versatility.

## 3. Method

### 3.1. Expectation Consistency Condition

Previous studies (Pampari & Ermon, 2020; Wang et al., 2020; Park et al., 2020; Wang et al., 2023; Hu et al., 2024) have empirically demonstrated that covariate shift can cause the confidence calibrated on the source domain to be uncalibrated on the target domain. However, empirical evidence alone cannot capture all possible scenarios. The theoretical underpinnings of these observations deserve to be explored to support this problem further and help solve it. To address this, this paper derives a necessary and sufficient condition for confidence calibration under covariate shift, as shown in Theorem 3.1.

**Theorem 3.1. (Expectation Consistency Condition)** $\forall 1 \leq k \leq K$, $P_s(Y_k = 1|S) = P_t(Y_k = 1|S)$ *if and only if:* $\mathbb{E}_{X \sim P_s(X|S)}[P(Y_k = 1|X)] = \mathbb{E}_{X \sim P_t(X|S)}[P(Y_k = 1|X)]$, *where* $P(Y_k = 1|X) = P_s(Y_k = 1|X) = P_t(Y_k = 1|X)$. *The proof is provided in Appendix B.*

**Remark on Theorem 3.1:** The source domain can usually be easily calibrated well using general calibration methods,

at least much better than the target domain (see Appendix C). Theorem 3.1 tells us that as long as *Expectation consistency condition* is met, the target domain can be calibrated as well as the source domain. Condition $\mathbb{E}_{X \sim P_s(X|S)}[P(Y_k = 1|X)] = \mathbb{E}_{X \sim P_t(X|S)}[P(Y_k = 1|X)]$ is strictly weaker than covariate distribution alignment (i.e., $P_s(X) = P_t(X)$), as it only requires equivalence in the expectations of the true posterior probability $P(Y_k = 1|X)$ *w.r.t.* the confidence score's level set distribution (i.e., $P_s(X|S)$ or $P_t(X|S)$), rather than matching the entire input distribution. For instance, even if $P_s(X)$ and $P_t(X)$ differ significantly, calibration may still hold if the model's expected accuracy conditioned on $S$ aligns across domains. This insight moves the focus from aligning global covariate distributions to enforcing local consistency in critical statistics, enabling more efficient calibration strategies under covariate shift.

**Extension of Theorem 3.1:** Theorem 3.1 can be naturally extended to top-label calibration and class-wise calibration (see Appendix D). Intuitively, this only requires replacing the confidence score vector $S$ in Theorem 3.1 with the predicted class confidence $\hat{S}$ or the confidence score vector's components $S_k$.

**An Example:** Fig. 1 shows an example of Theorem 3.1, where covariate shift occurs but calibration error remains unchanged. Take $S_1 = 0.75$ as an example for calculation:

$$
\begin{aligned}
&P_s\left(Y_1 = 1|S = (0.75, 0.25)\right) \\
&= \sum_{X \in \{-1,1\}} P(Y_1 = 1|X) P_s(X|S = (0.75, 0.25)) \\
&= \sum_{X \in \{-1,1\}} 0.5 \cdot P_s(X|S = (0.75, 0.25)) = 0.5.
\end{aligned}
\tag{2}
$$

Similarly, it is easy to calculate that $P_t(Y_1 = 1|S = (0.75, 0.25)) = 0.5 = P_s(Y_1 = 1|S = (0.75, 0.25))$. The same holds if 0.75 is replaced with other values because $\mathbb{E}_{X \sim P_s(X|S)}[P(Y_1 = 1|X)] = \mathbb{E}_{X \sim P_t(X|S)}[P(Y_1 =$

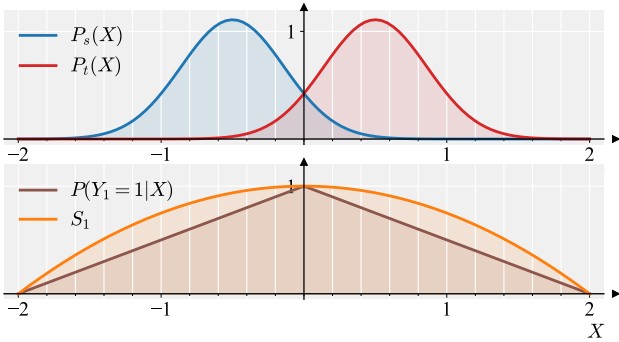

*Figure 1.* A binary classification example where covariate shift occurs but calibration error remains unchanged, where $P(Y|X) = (P(Y_1|X), P(Y_2|X))$ and $S = (S_1, S_2)$. $P(Y_2|X) = 1 - P(Y_1|X)$ and $S_2 = 1 - S_1$. $P_s(X) = (\sqrt{2\pi})^{-1} e^{-0.5(X+0.5)^2}$, $P_t(X) = (\sqrt{2\pi})^{-1} e^{-0.5(X-0.5)^2}$, $S_1 = -0.25X^2 + 1$, and $P(Y_1 = 1|X) = -0.5|X| + 1$.

$1|X)]$ holds for $\forall S_1 \in [0, 1]$. Moreover, such examples are infinite because they include but are not limited to all examples where $P(Y_1 = 1|X)$ or $S_1$ curves in Fig. 1 are symmetric *w.r.t.* the y-axis.

### 3.2. Expectation Consistency Loss

According to Theorem 3.1, *Expectation consistency condition* ensures that the target domain can be calibrated as effectively as the source domain. Specifically, in canonical calibration, *Expectation consistency condition* can be rewritten as follows:

$$\mathbb{E}_{P_t(S)} \left\| \mathbb{E}_{P_s(X|S)} P(Y|X) - \mathbb{E}_{P_t(X|S)} P(Y|X) \right\| = 0, \quad (3)$$

where $S = (S_1, \cdots, S_K) = (f_1(X), \cdots, f_K(X)) = f(X)$, $P_t(S)$ represents the probability density of the predicted confidence score vector on the target domain. Therefore, *Expectation consistency loss* can be naturally constructed as:

$$L_{ecl} = \mathbb{E}_{P_t(S)} \left\| \mathbb{E}_{P_s(X|S)} P(Y|X) - \mathbb{E}_{P_t(X|S)} P(Y|X) \right\|, \quad (4)$$

To estimate $P(Y|X)$ in practice, we train an additional classification head on the original classifier's backbone, where the label is the one-hot encoded $Y$ and the input data is $X$. This classification head can be trained end-to-end with the original classifier (freeze the backbone when training this classification head). Optionally, this classification head can also be calibrated on the source domain.

**Extension of Expectation Consistency Loss:** Eq. 4 is *Expectation consistency loss* for canonical calibration. Similarly, *Expectation consistency loss* for class-wise and top-label calibration can be obtained (see Appendix E).

### 3.3. Empirical Calculation and Differentiability

$L_{ecl}$ can be empirically estimated using confidence binning and Monte Carlo sampling:

$$\begin{cases} \hat{L}_{ecl} = \sum_{j=1}^{B} \frac{\sharp b_j^{(t)}}{\sharp D_t} \left\| \hat{\mathbb{E}}_{s,j} - \hat{\mathbb{E}}_{t,j} \right\|, \\[2ex] \hat{\mathbb{E}}_{s,j} = \frac{1}{\sharp D_s^{(j)}} \sum_{x \in D_s^{(j)}} \hat{P}(Y = y|X = x), \\[2ex] \hat{\mathbb{E}}_{t,j} = \frac{1}{\sharp D_t^{(j)}} \sum_{x \in D_t^{(j)}} \hat{P}(Y = y|X = x), \end{cases} \quad (5)$$

where $B$ represents the number of bins, $b_j^{(t)}$ represents the $j$-th bin in the target domain, $\sharp b_j^{(t)}$ represents sample size of $b_j^{(t)}$, $\sharp D_t$ represents sample size of $D_t$, $D_s^{(j)}$ represents the level set of $b_j^{(t)}$ in the source domain, $D_t^{(j)}$ represents the level set of $b_j^{(t)}$ in the target domain, and $\hat{P}(Y = y|X = x)$ represents the observation of $P(Y|X)$.

**Differentiability:** The confidence binning operation in Eq. 5 is non-differentiable (Karandikar et al., 2021; Bohdal et al., 2023; Popordanoska et al., 2022), so it cannot be directly used for classifier training. Therefore, a differentiable version is proposed below. Specifically, we replace hard bin membership with a smooth anchor-based assignment over confidence bins. For canonical calibration, the $i$-th confidence vector $S^{(i)} \in \Delta_{K-1}$ is a point in simplex. We introduce $B$ anchor points $a_j \in \Delta_{K-1}$ and define for the soft assignment of the $i$-th confidence vector $S^{(i)}$:

$$\omega_{ij} = \frac{\exp(-\|S^{(i)} - a_j\|_2^2/\tau)}{\sum_{r=1}^{B} \exp(-\|S^{(i)} - a_r\|_2^2/\tau)}, \quad (6)$$

with temperature $\tau > 0$. Denoting $p^{(i)} = P(Y|X_i)$ as the output of the additional classification head (as described in Section 3.2), we obtain for each bin $j$ and domain $d \in \{s, t\}$:

$$\hat{\mathbb{E}}_{d,j} = \frac{\sum_i \omega_{ij}^d \, p^{(i)}}{\sum_i \omega_{ij}^d + \varepsilon}, \qquad n_j^d = \sum_i \omega_{ij}^d, \quad (7)$$

with a small stabilizer $\varepsilon > 0$, where $\omega_{ij}^d$ represents the soft assignment in domain $d$. Then, the differentiable ECL is:

$$\hat{L}_{ecl} = \sum_{j=1}^{B} w_j \left\| \hat{\mathbb{E}}_{s,j} - \hat{\mathbb{E}}_{t,j} \right\|, \qquad w_j = \frac{n_j^t}{\sum_{r=1}^{B} n_r^t}. \quad (8)$$

**Extension of Differentiable ECL:** Eq. 8 is differentiable *Expectation consistency loss* for canonical calibration. Similarly, differentiable *Expectation consistency loss* for top-label and class-wise calibration can be obtained (see Appendix F).

## 3.4. Sample Complexity Analysis

**Theorem 3.2. (Sample Complexity of ECL Estimation)**
*Let $\varepsilon \in (0,1)$ and $\delta \in (0,1)$. Consider the empirical ECL in Eq. 5 (or Eq. 8) with $B$ bins, bin weights $w_j$ (target-domain proportions or their soft analogs), and per-bin sample counts $n_j^t$ and $n_j^s$. There exist absolute constants $C > 0$ such that, with probability at least $1 - \delta$,*

$$\left| \hat{L}_{ecl} - L_{ecl} \right| \le C \sqrt{\log\left(\frac{2BK}{\delta}\right) \sum_{j=1}^{B} w_j \left(\frac{1}{n_j^t} + \frac{1}{n_j^s}\right)}. \quad (9)$$

*Its proof is provided in Appendix G.*

**Remark on Theorem 3.2:** Theorem 3.2 implies ECL has a similar sample complexity as histogram binning for ECE, namely $\mathcal{O}(B/\varepsilon^2)$, and the weights $w_j$ explicitly cap the influence of sparse bins. This sample complexity is also similar to that of some point estimation methods (e.g., maximum likelihood estimation with $\mathcal{O}(1/\varepsilon^2)$) and is feasible for most real-world learning tasks.

## 3.5. Mini-Batch Trainability

Most modern deep learning methods are trained using mini-batches, where a small subset of data is processed at each step to compute the loss and update the model via gradient descent. This poses a challenge for confidence calibration loss, since small sample batches often fail to provide sufficiently accurate estimates of calibration error. Similar to the widely used cross-entropy loss, mini-batch trainability requires that the gradient computed on a mini-batch be an unbiased estimate of the gradient over the entire dataset, i.e., $E_{D_s^m, D_t^m}\left[\nabla_\theta \hat{L}_{ecl}^m\right] = \nabla_\theta \hat{L}_{ecl}$, where $D_s^m$ and $D_t^m$ represent mini-batches from the source and target domains, respectively. Therefore, we propose an equivalent formulation of Eq. 8 and prove its mini-batch trainability, as established in Theorem 3.3.

**Theorem 3.3. (ECL Mini-Batch Trainability)** *Eq. 10 is asymptotically equivalent to Eq. 8, and it satisfies $E_{D_s^m, D_t^m}\left[\nabla_\theta \hat{L}_{ecl}^{mini}\right] = \nabla_\theta \hat{L}_{ecl}$, and its proof is provided in Appendix H:*

$$\hat{L}_{ecl}(\theta, u_j^s, u_j^t) = \sum_{j=1}^{B} w_j \|u_j^s - u_j^t\|$$
$$+ \sum_{j=1}^{B} \sum_{i \in D_s} \omega_{i,j}^s \|u_j^s - p^{(i)}(\theta)\|^2 \quad (10)$$
$$+ \sum_{j=1}^{B} \sum_{i \in D_t} \omega_{i,j}^t \|u_j^t - p^{(i)}(\theta)\|^2,$$

*where $u_j^s$ and $u_j^t$ are learnable parameters used to approximate $\hat{\mathbb{E}}_{s,j}$ and $\hat{\mathbb{E}}_{t,j}$ during the training process, and $p^{(i)}(\theta)$*

---

**Algorithm 1** ECL Mini-Batch Training.

1: **Input:**
2:   bins $j = 1 \dots B$, hyperparameters $\alpha_{ema}, N_{prox}, \lambda$
3:   $u_j^s = \mathbf{0} \in \mathbb{R}^K, \forall j; u_j^t = \mathbf{0} \in \mathbb{R}^K, \forall j$
4: **for** each iteration **do**
5:   Sample mini-batches $D_s^m, D_t^m$;
6:   Compute weights $\omega_{ij}^s, \omega_{ij}^t$;
7:   $n_{s,j} \leftarrow \sum_{i \in D_s^m} \omega_{ij}^s, m_{s,j} \leftarrow \sum_{i \in D_s^m} \omega_{ij}^s p^{(i)}(\theta)$;
8:   $n_{t,j} \leftarrow \sum_{i \in D_t^m} \omega_{ij}^t, m_{t,j} \leftarrow \sum_{i \in D_t^m} \omega_{ij}^t p^{(i)}(\theta)$;
9:   $w_j \leftarrow n_{t,j} / \sum_{r=1}^{B} n_{t,r}; L_{ecl} \leftarrow 0$;
10:   **for** each bin $j$ **do**
11:     $u_s, u_t \leftarrow$ cached $u_j^s, u_j^t$
12:     **for** $i = 1$ to $N_{prox}$ **do**
13:       $v_s \leftarrow (m_{s,j}/n_{s,j}) - u_t, \tau_s = \dfrac{w_j}{2n_{s,j}}$
14:       $u_s \leftarrow u_t + \text{shrink}(v_s, \tau_s)$
15:       $v_t \leftarrow (m_{t,j}/n_{t,j}) - u_s, \tau_t = \dfrac{w_j}{2n_{t,j}}$
16:       $u_t \leftarrow u_s + \text{shrink}(v_t, \tau_t)$
17:     **end for**
18:     $\tilde{u}_j^s, \tilde{u}_j^t \leftarrow u_s.\text{detach}(), u_t.\text{detach}()$
19:     $u_j^s \leftarrow (1 - \alpha_{ema})u_j^s + \alpha_{ema}\tilde{u}_j^s$
20:     $u_j^t \leftarrow (1 - \alpha_{ema})u_j^t + \alpha_{ema}\tilde{u}_j^t$
21:     $L_{ecl} += \sum_{i \in D_s^m} \omega_{ij}^s \|\tilde{u}_j^s - p^{(i)}(\theta)\|^2$
22:     $L_{ecl} += \sum_{i \in D_t^m} \omega_{ij}^t \|\tilde{u}_j^t - p^{(i)}(\theta)\|^2$
23:   **end for**
24:   Compute the cross-entropy loss $L_{ce}$
25:   Backpropagate $L_{ce} + \lambda L_{ecl}$ and update $\theta$.
26: **end for**
27: **Return:** $\theta$

---

*denotes $P(Y|X_i)$ estimated by an additional classification head trained on the original classifier's backbone.*

**Remark on Theorem 3.3:** Because nonlinear operators such as norms do not commute with expectations, computing Eq. 8 directly on a mini-batch introduces bias into the gradient, as demonstrated in the proof of Theorem 3.3. By introducing auxiliary variables ($u_j^s$ and $u_j^t$) for learning the expectation over the full dataset, Eq. 10 perfectly avoids this problem. Algorithm 1 provides the pseudocode for the actual calculation of Eq. 10. Specifically, $u_j^s$ and $u_j^t$ in Eq. 10 can be solved using alternating proximal updates (Bolte et al., 2014), as detailed in Algorithm 1.

**Extension of ECL Mini-Batch Training:** Algorithm 1 is ECL mini-batch training for canonical calibration. Similarly, ECL mini-batch training for top-label and class-wise calibration can be obtained (see Appendix I).

# 4. Results

The effectiveness of the proposed method is verified from two perspectives: 1) Verify calibration effectiveness on simulated covariate shift data; 2) Comparison with state-of-the-art calibration methods on real-world covariate shift datasets.

## 4.1. Calibration on Simulated Covariate Shift Data

**Experimental Setup:** To observe covariate shift, we model source and target domain covariates as normal and uniform distributions (Figs. 2(a-b) and Figs. 3(a-b), respectively). For normal distributions, source domain has mean [0, 0] and covariance [[5, 0], [0, 5]], while target domain has mean [2, 2] and the same covariance. For uniform distributions, source domain is 2D uniform on $[-2.5, 2.5]^2$ and target domain on $[-1.5, 3.5]^2$. Since $P_s(Y|X) = P_t(Y|X)$, the labeling function is identical in both domains, shown by the blue segmentation curves in Figs. 2(a-b) and Figs. 3(a-b). We sample 400 points from each domain. The classifier is a three-layer backpropagation neural network trained with Adam optimizer (learning rate 0.001) for 100 epochs. Reliability diagrams use 15 bins (Guo et al., 2017; Zhang et al., 2020). The classification head estimating $P(Y|X)$ (or $P(Y^* = \hat{Y}|X)$ for top-label calibration) is calibrated on the source domain using Soft-ECE loss.

**Results:** Fig. 2 and Fig. 3 (in Appendix J.1) show the calibration results on the simulated covariate shift dataset. Fig. 2 shows the case where the covariate distribution is normally distributed, and Fig. 3 shows the case where the covariate distribution is uniformly distributed. The ECL's results shown in the different reliability diagrams are from different ECL versions about different calibration paradigms. In the reliability diagrams, the outputs of the top-label and class-wise reliability diagrams after ECL calibration are closer to the diagonal, indicating improved calibration performance. In canonical calibration reliability diagrams, high calibration errors usually occur near the midpoint of a side of the large triangle, corresponding to situations where the confidence scores of each component of the predicted vector are not very high. Overall, the number of highlighted small triangle bins after ECL calibration in canonical reliability diagrams will decrease (see Fig. 2) or the color will become dark blue (see Fig. 3). From evaluating metrics under the two covariate distribution shifts, ECL can stably reduce calibration error in all three calibration paradigms and improve accuracy in most cases.

## 4.2. Calibration on Real-World Covariate Shift Datasets

### 4.2.1. EXPERIMENTAL SETUP

**Datasets and Networks:** To reflect the effectiveness of calibration methods on the real-world dataset, three differ-ent types of covariate shift datasets are selected for experiments: 1) Digit recognition dataset includes three different domains (MNIST (Lecun et al., 1998), USPS (Hull, 1994), and SVHN (Netzer et al., 2011)); 2) a domain adaptation dataset PACS contains four different domains (Photo, Art Painting, Cartoon, and Sketch) (Li et al., 2017); 3) a large-scale dataset ImageNet-Sketch with 1000 classes contains two domains (ImageNet and Sketch) (Wang et al., 2019). When constructing covariate shift datasets, one domain of the dataset is used as the target domain, and the other domains are merged into the source domain. The commonly used networks on these datasets are used in the experiments, i.e., LeNet (Lecun et al., 1998), ResNet (He et al., 2016), DenseNet (Huang et al., 2017), Wide-ResNet (Zagoruyko & Komodakis, 2016) and ViT (Dosovitskiy et al., 2021).

**Calibration Metrics:** To comprehensively evaluate the calibration performance in three calibration paradigms, we used the following calibration metrics to evaluate the calibration methods: 1) **ECE**: The classic expected calibration error (Guo et al., 2017) for top-label calibration; 2) **CwECE**: Class-wise expected calibration error (Kull et al., 2019) for class-wise calibration; 3) **ECE$^{KDE}$**: a consistent and differentiable canonical calibration metric for canonical calibration. In addition, we report $\Delta$ACC as the accuracy change relative to the uncalibrated classi-fier under the same task/architecture, defined as $\Delta$ACC = ACC(ECL) − ACC(Uncal).

**Baselines:** For a comprehensive comparison, the follow-ing methods are compared: 1) **Uncal**: Training using only cross-entropy loss; 2) **Soft-ECE** (Karandikar et al., 2021): A softened differentiable ECE loss; 3) **DECE** (Bohdal et al., 2023): Another softened differentiable ECE loss; 4) **KDE** (Popordanoska et al., 2022): a differentiable canonical cal-ibration loss; 5) **TS** (Guo et al., 2017): Classic post-hoc calibration method with temperature scaling; 6) **TransCal** (Wang et al., 2020): a debiasing calibration method based on importance weighting; 7) **DRL** (Wang et al., 2023): a calibration method based on distributionally robust learning; 8) **PseudoCal** (Hu et al., 2024): a calibration method based on mixup data synthesis; 9) **Oracle**: Soft-ECE calibration using labels on the target domain.

### 4.2.2. RESULTS

Table 2 reports the calibration metric ECE for top-label calibration on the digit recognition benchmarks. Overall, ECL achieves the lowest (or near-lowest) ECE in most trans-fer tasks and network architectures, demonstrating strong calibration performance compared to state-of-the-art base-lines. The advantage of ECL is particularly evident on the SVHN dataset, which involves larger distribution shifts; for instance, on LeNet-5, ECL reduces the ECE from 61.9% (Uncalibrated) to 21.5%, substantially improving upon most

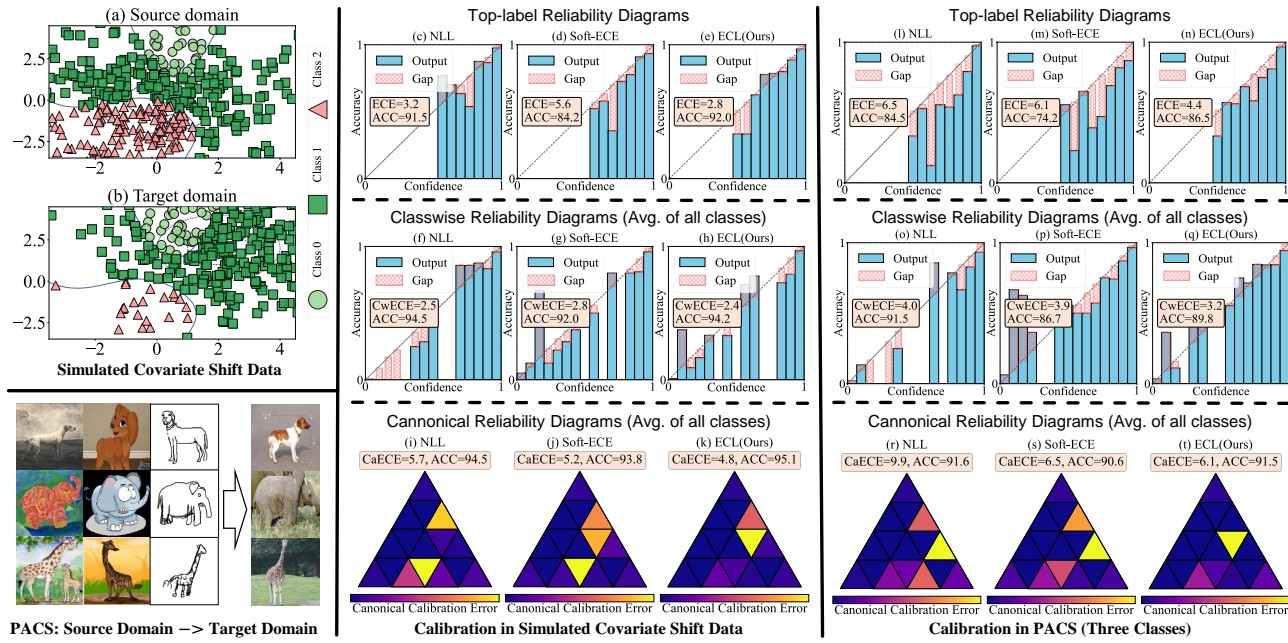

*Figure 2.* Calibration effect display. Figures (c) to (k) show the calibration effect on the simulated covariate shift dataset (see Figures (a) and (b)), and Figures (l) to (t) show the calibration effect on the real-world covariate shift dataset PACS (three classes). NLL represents cross-entropy loss, Soft-ECE represents softened differentiable ECE loss, CwECE represents class-wise ECE, and CaECE represents canonical ECE. Results from the three types of reliability diagrams and calibration metrics demonstrate that our method preserves or improves classifier accuracy while substantially reducing calibration errors. Our code is available at https://github.com/NeuroDong/ECL.

baselines (e.g., PseudoCal at 52.4%). Furthermore, the ∆ACC values suggest that ECL often improves calibration while largely preserving the discriminative power of the classifier.

Extended results covering broader benchmarks and calibration paradigms are detailed in Appendices J.2, J.3, and J.4. Appendix J.2 provides additional top-label calibration results on the PACS and ImageNet-Sketch datasets. Appendices J.3 and J.4 present comprehensive evaluations for class-wise and canonical calibration, respectively, across all three dataset suites (Digit, PACS, and ImageNet-Sketch). Overall, these experiments show that ECL is highly competitive and frequently achieves the lowest errors in terms of ECE, CwECE, and $ECE^{KDE}$.

### 4.3. Ablation

**Mini-Batch Trainability:** We empirically verify the role of our proposed mini-batch training strategy by comparing it with a naive baseline, **Mini-Batch Non-Trainable ECL**, which directly computes Eq. 8 on mini-batches. As shown in Table 7 (see Appendix), our **Mini-Batch Trainable ECL** (Algorithm 1) is more stable and achieves better calibration in most settings, supporting the effectiveness of the auxiliary variable formulation (Theorem 3.3).

**Loss Weight $\lambda$:** To balance the cross-entropy loss and ECL,

we employ an adaptive weighting strategy: $\lambda = \beta^\gamma$, where $\beta = \left( \sum_i \mathcal{L}_{ce}^{(i)} \right) / \left( \sum_i \mathcal{L}_{ecl}^{(i)} \right)$ acts as a balancing factor between the two loss magnitudes. The hyperparameter $\gamma$ controls the sensitivity of this regularization. Our ablation study in Table 8 (see Appendix) suggests that a linear scaling ($\gamma = 1.0$) provides a strong trade-off between calibration improvement and accuracy preservation in our tested settings.

## 5. Discussion

**Why It Works:** The essence of ECL is to reorganize the confidence space rather than aligning covariate distributions. For each confidence level $S$, it ensures that source and target samples achieving this confidence share the same expected true posterior $P(Y|X)$, effectively grouping samples with similar true accuracy into the same confidence bins regardless of their input distributions. This level set alignment method directly addresses the essential need for confidence calibration under covariate shift, thereby achieving stable and effective calibration.

**Potential Impact, Limitations, and Future Work:** We rethink confidence calibration under covariate shifts by moving beyond traditional importance weighting. Our findings reveal that strict covariate distribution alignment is unnecessary; instead, a weaker condition—the *Expectation Consis-*

*Table 2.* ECE (%) for top-label calibration on digit recognition datasets. The reported results represent the mean and standard deviation derived from ten runs.

| Datasets | Uncal | Soft-ECE | DECE | KDE | ECE ↓ TS | TransCal | DRL | PseudoCal | ECL (Ours) | Oracle ↓ | ΔACC(%) |
|---|---|---|---|---|---|---|---|---|---|---|---|
| **→ MNIST** | | | | | | | | | | | |
| LeNet-5 | $27.3_{\pm 2.63}$ | $27.8_{\pm 2.15}$ | $26.5_{\pm 1.88}$ | $27.9_{\pm 2.01}$ | $27.7_{\pm 1.34}$ | $26.9_{\pm 1.16}$ | $22.3_{\pm 2.04}$ | $9.08_{\pm 0.71}$ | $8.52_{\pm 0.78}$ | $0.30_{\pm 0.01}$ | $-0.92_{\pm 0.35}$ |
| ResNet20 | $16.2_{\pm 1.51}$ | $16.5_{\pm 1.22}$ | $15.8_{\pm 1.45}$ | $16.1_{\pm 1.10}$ | $15.3_{\pm 1.04}$ | $13.1_{\pm 0.99}$ | $10.2_{\pm 0.72}$ | $8.22_{\pm 0.53}$ | $7.88_{\pm 0.45}$ | $1.54_{\pm 0.04}$ | $+1.25_{\pm 0.42}$ |
| DenseNet40 | $23.4_{\pm 1.79}$ | $23.6_{\pm 1.55}$ | $22.1_{\pm 1.62}$ | $22.9_{\pm 1.48}$ | $21.6_{\pm 1.71}$ | $19.8_{\pm 0.96}$ | $14.8_{\pm 0.95}$ | $9.72_{\pm 0.68}$ | $9.15_{\pm 0.61}$ | $1.40_{\pm 0.03}$ | $+0.68_{\pm 0.20}$ |
| **→ USPS** | | | | | | | | | | | |
| LeNet-5 | $22.9_{\pm 1.50}$ | $23.1_{\pm 1.28}$ | $22.4_{\pm 1.40}$ | $22.8_{\pm 1.35}$ | $22.7_{\pm 1.13}$ | $21.8_{\pm 1.32}$ | $15.5_{\pm 1.16}$ | $8.92_{\pm 0.45}$ | $8.12_{\pm 0.42}$ | $1.54_{\pm 0.02}$ | $-0.85_{\pm 0.25}$ |
| ResNet20 | $9.14_{\pm 0.74}$ | $9.32_{\pm 0.65}$ | $9.05_{\pm 0.71}$ | $9.45_{\pm 0.55}$ | $9.12_{\pm 0.84}$ | $8.36_{\pm 0.45}$ | $7.99_{\pm 0.66}$ | $5.01_{\pm 0.30}$ | $5.25_{\pm 0.28}$ | $2.23_{\pm 0.06}$ | $+1.42_{\pm 0.37}$ |
| DenseNet40 | $15.7_{\pm 0.83}$ | $15.9_{\pm 0.76}$ | $15.3_{\pm 0.92}$ | $15.8_{\pm 1.01}$ | $13.1_{\pm 1.02}$ | $12.1_{\pm 1.04}$ | $7.92_{\pm 0.47}$ | $5.34_{\pm 0.34}$ | $4.96_{\pm 0.28}$ | $2.54_{\pm 0.05}$ | $-0.76_{\pm 0.18}$ |
| **→ SVHN** | | | | | | | | | | | |
| LeNet-5 | $61.9_{\pm 6.16}$ | $62.2_{\pm 5.50}$ | $60.8_{\pm 5.22}$ | $62.5_{\pm 5.80}$ | $61.3_{\pm 5.89}$ | $63.7_{\pm 4.94}$ | $23.7_{\pm 1.93}$ | $52.4_{\pm 4.55}$ | $21.5_{\pm 1.51}$ | $1.03_{\pm 0.02}$ | $+1.65_{\pm 0.65}$ |
| ResNet20 | $68.2_{\pm 6.44}$ | $67.5_{\pm 5.92}$ | $66.9_{\pm 6.10}$ | $67.8_{\pm 6.25}$ | $68.1_{\pm 6.13}$ | $59.4_{\pm 4.63}$ | $40.1_{\pm 3.77}$ | $48.2_{\pm 3.95}$ | $36.8_{\pm 2.08}$ | $0.50_{\pm 0.02}$ | $+2.12_{\pm 0.88}$ |
| DenseNet40 | $80.8_{\pm 6.26}$ | $81.2_{\pm 5.88}$ | $79.5_{\pm 6.05}$ | $81.1_{\pm 6.15}$ | $77.2_{\pm 6.98}$ | $72.9_{\pm 5.13}$ | $42.0_{\pm 3.36}$ | $64.7_{\pm 4.72}$ | $38.4_{\pm 3.21}$ | $0.86_{\pm 0.03}$ | $-1.15_{\pm 0.45}$ |

(Row group label: **Digit**)

*tency Condition*—is sufficient for target domain calibration. This insight has the potential to inspire further research and enhance decision-making in safety-critical cross-population applications. However, our method assumes invariant posterior class probabilities ($P(Y|X)$), a common assumption among other methods in this field. Consequently, scenarios involving label shift, where the input-output relationship changes, fall outside the scope of this work. Future work will explore extending our framework to address calibration under both covariate and label shifts.

## 6. Conclusion

This paper rethinks confidence calibration under covariate shifts by moving beyond the traditional importance weighting paradigm. We derive a necessary and sufficient condition for confidence calibration under covariate shifts, termed the *Expectation Consistency Condition*, which reveals that covariate shifts do not necessarily lead to uncalibrated confidence and provides a weaker condition than global covariate distribution alignment. Building upon this theoretical foundation, we propose the *Expectation Consistency Loss* (ECL), an unsupervised domain adaptation loss that can be seamlessly applied to canonical, class-wise, and top-label calibration paradigms. Furthermore, we prove that ECL shares the same sample complexity as histogram binning for ECE estimation and provide a theoretically grounded mini-batch training scheme that enables unbiased gradient computation. Extensive experiments on both simulated and real-world covariate shift datasets demonstrate that ECL achieves competitive calibration errors across all three calibration paradigms while generally preserving classifier accuracy. Our work opens new avenues for confidence calibration research by shifting the focus from global distribution alignment to enforcing local consistency in critical statistics.

## Acknowledgements

This work was supported by the Science and Technology Innovation Program of Hunan Province (Grant Number: 2024RC1007) and the Central South University Post-Graduate Independent Exploration and Innovation Project (Grant Number: 2025ZZTS0616).

## Impact Statement

This paper presents work whose goal is to advance the field of Machine Learning. There are many potential societal consequences of our work, none which we feel must be specifically highlighted here.

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

# Appendix

## A. Top-label Calibration and Class-wise Calibration

**Definition A.1.** **(Top-label Calibration)** A classifier is perfectly top-label calibrated if the following equation holds:

$$P(Y^* = \hat{Y}|\hat{S} = \hat{s}) = \hat{s}, \tag{11}$$

where $Y^* = \operatorname{argmax}_k\{Y_k\}_{1 \leq k \leq K}$ is the true class scalar, $\hat{Y} = \operatorname{argmax}_k\{S_k\}_{1 \leq k \leq K}$ is the predicted class, $\hat{S} = \max\{S_k\}_{1 \leq k \leq K}$ is the confidence score of the predicted class, and $\hat{s}$ is the observed value on $\hat{S}$.

**Definition A.2.** **(Class-wise Calibration)** A classifier is perfectly class-wise calibrated if the following equation holds:

$$P(Y_k = 1|S_k = s_k) = s_k, \forall 1 \leq k \leq K, \tag{12}$$

where $Y_k$ is the $k$-th component of the one-hot label $Y$, and $S_k$ is the $k$-th component of confidence score vector $S$, and $s_k$ is the observed value on $S_k$.

## B. Proof of Theorem 3.1

*Proof.* First, according to Total Probability Theorem, the following holds:

$$\begin{aligned} P(Y_k = 1 \mid S) &= \int_X P(Y_k = 1, X \mid S)\, dX \\ &= \int_X P(Y_k = 1 \mid X, S)\, P(X \mid S)\, dX \\ &= \int_X P(Y_k = 1 \mid X)\, P(X \mid S)\, dX \\ &= \mathbb{E}_{X \sim P(X|S)}[P(Y_k = 1 \mid X)]. \end{aligned} \tag{13}$$

where the second-to-last equality is because $X$ contains all the information that $S$ can provide. According to the definition of covariate shift, $P_s(Y_k = 1|X) = P_t(Y_k = 1|X)$. Therefore, if $P_s(Y_k = 1|S) = P_t(Y_k = 1|S)$, then:

$$\mathbb{E}_{X \sim P_s(X|S)}[P(Y_k = 1|X)] = \mathbb{E}_{X \sim P_t(X|S)}[P(Y_k = 1|X)]. \tag{14}$$

where $P(Y_k = 1|X) = P_s(Y_k = 1|X) = P_t(Y_k = 1|X)$. Conversely, if $\mathbb{E}_{X \sim P_s(X|S)}[P(Y_k = 1|X)] = \mathbb{E}_{X \sim P_t(X|S)}[P(Y_k = 1|X)]$, it also holds that $P_s(Y_k = 1|S) = P_t(Y_k = 1|S)$. □

## C. Calibration Comparison Between Source and Target Domains

Typically, a classifier's calibration error in the source domain is significantly lower than that in the target domain because there is no distribution shift that leads to insufficient generalization. For the sake of rigor, we still verified this natural assumption through experiments. Table 3 presents the experimental results. We use soft-ECE as the calibration method to calibrate the models in the source domain. All three calibration metrics for different calibration paradigms show that the calibration error in the source domain is significantly lower than that in the target domain. Therefore, even just making the calibration error in the target domain as good as that in the source domain would be a significant improvement.

*Table 3.* Comparison of calibration errors between the source and target domains. The subscript $s$ denotes the source domain, while the subscript $t$ denotes the target domain. ResNet-20 is used for the Digit dataset, ResNet-50 for the PACS dataset, and ViT-L for the ImageNet-Sketch dataset.

| Dataset | $\text{ECE}_s$ | $\text{ECE}_t$ | $\text{CwECE}_s$ | $\text{CwECE}_t$ | $\text{ECE}_s^{KDE}$ | $\text{ECE}_t^{KDE}$ |
|---|---|---|---|---|---|---|
| Digit (USPS + SVHN → MNIST) | $1.54_{\pm0.04}$ | $16.2_{\pm1.51}$ | $0.39_{\pm0.01}$ | $3.14_{\pm0.31}$ | $0.39_{\pm0.02}$ | $2.97_{\pm0.23}$ |
| PACS (Art + Cartoon + Sketch → Photo) | $3.84_{\pm0.23}$ | $22.3_{\pm2.16}$ | $0.58_{\pm0.01}$ | $7.87_{\pm0.31}$ | $0.42_{\pm0.04}$ | $7.58_{\pm0.37}$ |
| ImageNet-Sketch (ImageNet → Sketch) | $1.47_{\pm0.11}$ | $55.8_{\pm4.34}$ | $0.93_{\pm0.09}$ | $12.7_{\pm0.87}$ | $0.86_{\pm0.06}$ | $12.3_{\pm0.73}$ |

## D. Extension of Theorem 3.1

**Theorem D.1.** **(Expectation Consistency Condition for Top-label Calibration)** $P_s(Y^* = \hat{Y}|\hat{S}) = P_t(Y^* = \hat{Y}|\hat{S})$ *if and only if:* $\mathbb{E}_{X \sim P_s(X|\hat{S})}[P(Y^* = \hat{Y}|X)] = \mathbb{E}_{X \sim P_t(X|\hat{S})}[P(Y^* = \hat{Y}|X)]$, *where* $P(Y^* = \hat{Y}|X) = P_s(Y^* = \hat{Y}|X) = P_t(Y^* = \hat{Y}|X)$.

*Proof.* First, according to Total Probability Theorem, the following holds:

$$
\begin{aligned}
P(Y^* = \hat{Y}|\hat{S}) &= \int_X P(Y^* = \hat{Y}, X|\hat{S})dX \\
&= \int_X P(Y^* = \hat{Y}|X, \hat{S})P(X|\hat{S})dX = \int_X P(Y^* = \hat{Y}|X)P(X|\hat{S})dX,
\end{aligned}
\tag{15}
$$

where the last equality is because $X$ contains all the information that $\hat{S}$ can provide. According to the definition of covariate shift, $P_s(Y^*|X) = P_t(Y^*|X)$. Because the source domain and the target domain share a fixed classifier, $P_s(\hat{Y}|X) = P_t(\hat{Y}|X)$. Then, it holds:

$$
\begin{aligned}
P_s(Y^*, \hat{Y}|X) &= P_s(Y^*|\hat{Y}, X)P_s(\hat{Y}|X) \\
&= P_s(Y^*|X)P_s(\hat{Y}|X) = P_t(Y^*|X)P_t(\hat{Y}|X) = P_t(Y^*, \hat{Y}|X).
\end{aligned}
\tag{16}
$$

where the third equality is because the classifier is fixed, $\hat{Y}$ is a deterministic function of $X$. Therefore, $P_s(Y^* = \hat{Y}|X) = P_t(Y^* = \hat{Y}|X)$. According to Eq. 15, if $P_s(Y^* = \hat{Y}|\hat{S}) = P_t(Y^* = \hat{Y}|\hat{S})$, then $\mathbb{E}_{X \sim P_s(X|\hat{S})}[P(Y^* = \hat{Y}|X)] = \mathbb{E}_{X \sim P_t(X|\hat{S})}[P(Y^* = \hat{Y}|X)]$. Conversely, if $\mathbb{E}_{X \sim P_s(X|\hat{S})}[P(Y^* = \hat{Y}|X)] = \mathbb{E}_{X \sim P_t(X|\hat{S})}[P(Y^* = \hat{Y}|X)]$, it also holds that $P_s(Y^* = \hat{Y}|\hat{S}) = P_t(Y^* = \hat{Y}|\hat{S})$. $\square$

**Theorem D.2.** **(Expectation Consistency Condition for Class-wise Calibration)** $\forall 1 \le k \le K$, $P_s(Y_k = 1|S_k) = P_t(Y_k = 1|S_k)$ *if and only if* $\mathbb{E}_{X \sim P_s(X|S_k)}[P(Y_k = 1|X)] = \mathbb{E}_{X \sim P_t(X|S_k)}[P(Y_k = 1|X)]$, *where* $P(Y_k = 1|X) = P_s(Y_k = 1|X) = P_t(Y_k = 1|X)$.

*Proof.* By the law of total probability,

$$
P(Y_k = 1|S_k) = \int_X P(Y_k = 1, X|S_k)dX = \int_X P(Y_k = 1|X, S_k)P(X|S_k)dX = \int_X P(Y_k = 1|X)P(X|S_k)dX,
\tag{17}
$$

where the last step uses that $X$ contains all information in $S_k$ relevant to $Y_k$.

Under covariate shift $P_s(Y_k = 1|X) = P_t(Y_k = 1|X)$. Hence:

$$
P_s(Y_k = 1|S_k) = \int_X P(Y_k = 1|X)P_s(X|S_k)dX, \quad P_t(Y_k = 1|S_k) = \int_X P(Y_k = 1|X)P_t(X|S_k)dX.
\tag{18}
$$

Therefore $P_s(Y_k = 1|S_k) = P_t(Y_k = 1|S_k)$ iff:

$$
\int_X P(Y_k = 1|X)P_s(X|S_k)dX = \int_X P(Y_k = 1|X)P_t(X|S_k)dX,
\tag{19}
$$

which is exactly the desired expectation condition. $\square$

## E. Extension of Expectation Consistency Loss

### E.1. Expectation Consistency Loss for Top-Label Calibration

Recall the predicted class $\hat{Y} = \arg\max_k \{S_k\}_{1 \le k \le K}$, its confidence $\hat{S} = \max\{S_k\}_{1 \le k \le K}$, and the true class $Y^*$. Theorem D.1 states that preservation of top-label calibration across domains is equivalent to the expectation consistency condition:

$$
\mathbb{E}_{X \sim P_s(X|\hat{S})}[P(Y^* = \hat{Y}|X)] = \mathbb{E}_{X \sim P_t(X|\hat{S})}[P(Y^* = \hat{Y}|X)].
\tag{20}
$$

Therefore, *Expectation consistency loss* for top-label calibration can be naturally constructed as:

$$L_{ecl}^{top} = \mathbb{E}_{P_t(\hat{S})}\left|\mathbb{E}_{P_s(X|\hat{S})}P(Y^* = \hat{Y}|X) - \mathbb{E}_{P_t(X|\hat{S})}P(Y^* = \hat{Y}|X)\right|. \tag{21}$$

To estimate $P(Y^* = \hat{Y}|X)$ in practice, we train a binary classifier where the label is $1_{Y^*=\hat{Y}}$ and the input data is $X$. This binary classifier can be added to the original classifier as a classification head and trained end-to-end with the original classifier (freeze the backbone when training this classification head). Optionally, this binary classification head can also be calibrated on the source domain to obtain a more reliable estimate of $P(Y^* = \hat{Y}|X)$.

### E.2. Expectation Consistency Loss for Class-wise Calibration

For class-wise calibration, each coordinate $S_k$ must match $P(Y_k = 1|S_k)$. Theorem D.2 implies expectation consistency per class:

$$\mathbb{E}_{X\sim P_s(X|S_k)}[P(Y_k = 1|X)] = \mathbb{E}_{X\sim P_t(X|S_k)}[P(Y_k = 1|X)], \quad \forall k \in \{1, \ldots, K\}. \tag{22}$$

Therefore, *Expectation consistency loss* for class-wise calibration can be naturally constructed as:

$$L_{ecl}^{cw} = \sum_{k=1}^{K}\left[\mathbb{E}_{P_t(S_k)}\left|\mathbb{E}_{P_s(X|S_k)}P(Y_k = 1|X) - \mathbb{E}_{P_t(X|S_k)}P(Y_k = 1|X)\right|\right]. \tag{23}$$

To estimate $P(Y_k = 1|X)$ in practice, we train an additional classification head on the original classifier's backbone, where the label is $Y_k$ (the $k$-th component of the one-hot encoded label) and the input data is $X$. This classification head can be trained end-to-end with the original classifier (freeze the backbone when training this classification head). Optionally, this classification head can also be calibrated on the source domain.

## F. Extensions on Empirical Calculation and Differentiability

**Empirical Calculation and Differentiability for Top-label Calibration:** For top-label calibration, *Expectation Consistency Loss* can be empirically estimated using confidence binning and Monte Carlo sampling:

$$\begin{cases} \hat{L}_{ecl}^{top} = \sum_{j=1}^{B} \frac{\sharp b_j^{(t)}}{\sharp D_t}\left\|\hat{\mathbb{E}}_{s,j} - \hat{\mathbb{E}}_{t,j}\right\|, \\ \hat{\mathbb{E}}_{s,j} = \frac{1}{\sharp D_s^{(j)}}\sum_{x\in D_s^{(j)}}\hat{P}(Y^* = \hat{Y}|X = x), \\ \hat{\mathbb{E}}_{t,j} = \frac{1}{\sharp D_t^{(j)}}\sum_{x\in D_t^{(j)}}\hat{P}(Y^* = \hat{Y}|X = x), \end{cases} \tag{24}$$

where $B$ represents the number of bins, $b_j^{(t)}$ represents the $j$-th bin in the target domain, $\sharp b_j^{(t)}$ represents sample size of $b_j^{(t)}$, $\sharp D_t$ represents sample size of $D_t$, $D_s^{(j)}$ represents the level set of $b_j^{(t)}$ in the source domain, $D_t^{(j)}$ represents the level set of $b_j^{(t)}$ in the target domain, and $\hat{P}(Y^* = \hat{Y}|X = x)$ represents the observation of $P(Y^* = \hat{Y}|X)$. For differentiability, introduce anchors $a_j = (2j - 1)/(2B)$ and weights $\omega_{ij} = \exp(-(\hat{S}^{(i)} - a_j)^2/\tau)/\sum_r \exp(-(\hat{S}^{(i)} - a_r)^2/\tau)$ with temperature $\tau > 0$. Denoting $p^{(i)} = P(Y^* = \hat{Y}|X_i)$ as the output of the binary classification head (as described in Section E), we obtain for each bin $j$ and domain $d \in \{s, t\}$:

$$\hat{\mathbb{E}}_{d,j} = \frac{\sum_i \omega_{ij}^d p^{(i)}}{\sum_i \omega_{ij}^d + \varepsilon}. \tag{25}$$

Therefore, the differentiable ECL for top-label calibration is $\hat{L}_{ecl}^{top} = \sum_{j=1}^{B} w_j\|\hat{\mathbb{E}}_{s,j} - \hat{\mathbb{E}}_{t,j}\|$, where $w_j = \frac{\sum_i \omega_{ij}^t}{\sum_r \sum_i \omega_{ir}^t}$.

**Empirical Calculation and Differentiability for Class-wise Calibration:** For class-wise calibration, *Expectation*

*Consistency Loss* can be empirically estimated using confidence binning and Monte Carlo sampling:

$$
\begin{cases}
\hat{L}_{ecl}^{\mathrm{cw}} = \sum_{k=1}^{K} \sum_{j=1}^{B} \frac{\sharp b_{k,j}^{(t)}}{\sharp D_t} \left| \hat{\mathbb{E}}_{s,k,j} - \hat{\mathbb{E}}_{t,k,j} \right|, \\[2mm]
\hat{\mathbb{E}}_{s,k,j} = \frac{1}{\sharp D_{s,k}^{(j)}} \sum_{x \in D_{s,k}^{(j)}} \hat{P}(Y_k = 1 \mid X = x), \\[2mm]
\hat{\mathbb{E}}_{t,k,j} = \frac{1}{\sharp D_{t,k}^{(j)}} \sum_{x \in D_{t,k}^{(j)}} \hat{P}(Y_k = 1 \mid X = x).
\end{cases}
\tag{26}
$$

where $B$ is the number of bins per class, $b_{k,j}^{(t)}$ is the $j$-th bin for class-$k$ on the target domain (formed by binning $S_k$), $\sharp b_{k,j}^{(t)}$ is its size, $\sharp D_t$ is the target sample size, and $D_{s,k}^{(j)}, D_{t,k}^{(j)}$ are the level sets of $b_{k,j}^{(t)}$ on source/target domains, respectively. For differentiability, let anchors $a_j = \frac{2j-1}{2B}$ for $j = 1, \ldots, B$, and define soft weights for a sample $i$ with confidence $S_k^{(i)}$:

$$
\omega_{k,ij} = \frac{\exp\left( -(S_k^{(i)} - a_j)^2 / \tau \right)}{\sum_{r=1}^{B} \exp\left( -(S_k^{(i)} - a_r)^2 / \tau \right)}, \quad \tau > 0.
\tag{27}
$$

For domain $d \in \{s, t\}$, define

$$
\hat{\mathbb{E}}_{d,k,j} = \frac{\sum_i \omega_{k,ij}^d p_k^{(i)}}{\sum_i \omega_{k,ij}^d + \varepsilon}, \quad n_{k,j}^d = \sum_i \omega_{k,ij}^d, \quad p_k^{(i)} = P(Y_k = 1 | X_i),
\tag{28}
$$

with stabilizer $\varepsilon > 0$. The differentiable class-wise ECL becomes

$$
\hat{L}_{ecl}^{\mathrm{cw}} = \sum_{k=1}^{K} \sum_{j=1}^{B} w_{k,j} \left| \hat{\mathbb{E}}_{s,k,j} - \hat{\mathbb{E}}_{t,k,j} \right|, \quad w_{k,j} = \frac{n_{k,j}^t}{\sum_{r=1}^{B} n_{k,r}^t}.
\tag{29}
$$

## G. Proof of Theorem 3.2

*Proof.* For each bin $j$, define random variables $Z_{s,j} = \|\hat{\mathbb{E}}_{s,j} - \mathbb{E}_{P_s(X|S)} P(Y|X)\|$ and $Z_{t,j} = \|\hat{\mathbb{E}}_{t,j} - \mathbb{E}_{P_t(X|S)} P(Y|X)\|$. By the triangle inequality,

$$
\left| \hat{L}_{ecl} - L_{ecl} \right| \leq \sum_{j=1}^{B} w_j (Z_{s,j} + Z_{t,j}).
\tag{30}
$$

Using Hoeffding's inequality and a union bound over bins and classes, there exist absolute constants $C_1, C_2 > 0$ such that, with probability at least $1 - \delta$,

$$
Z_{s,j} \leq C_1 \sqrt{\frac{K \log(2BK/\delta)}{n_{s,j}}}, \quad Z_{t,j} \leq C_2 \sqrt{\frac{K \log(2BK/\delta)}{n_{t,j}}}, \quad \forall j = 1, \ldots, B.
\tag{31}
$$

Combining these bounds gives the desired result. $\qquad\square$

## H. Proof of Theorem 3.3

This proof proceeds in two steps. First we show that Eq. 10 is an auxiliary-variable reformulation of Eq. 8: minimizing the auxiliary variables $u_j^s, u_j^t$ in Eq. 10 recovers Eq. 8. Second we show that, under the auxiliary-variable formulation, the mini-batch gradient is an unbiased estimator of the full-sample gradient.

**Equivalence between Eq. 10 and Eq. 8.** Fix $\theta$ and consider minimizing the right-hand side of Eq. 10 with respect to the auxiliary vectors $u_j^s, u_j^t$ for each bin $j$. The terms that depend on $u_j^s, u_j^t$ are

$$
G_j(u_j^s, u_j^t) = w_j \|u_j^s - u_j^t\| + \sum_{i \in D_s} \omega_{i,j}^s \|u_j^s - p_i(\theta)\|^2 + \sum_{i \in D_t} \omega_{i,j}^t \|u_j^t - p_i(\theta)\|^2.
$$

Define the soft counts and weighted empirical means

$$n_j^s = \sum_{i \in D_s} \omega_{i,j}^s, \quad n_j^t = \sum_{i \in D_t} \omega_{i,j}^t, \quad \hat{\mathbb{E}}_{s,j} = \frac{1}{n_j^s} \sum_{i \in D_s} \omega_{i,j}^s p^{(i)}(\theta), \quad \hat{\mathbb{E}}_{t,j} = \frac{1}{n_j^t} \sum_{i \in D_t} \omega_{i,j}^t p^{(i)}(\theta).$$

The quadratic terms are strongly convex in $u_j^s, u_j^t$, so $G_j$ has a unique minimizer. Taking (sub)gradients $w.r.t.$ $u_j^s, u_j^t$ and setting them to zero yields

$$2n_j^s(u_j^s - \hat{\mathbb{E}}_{s,j}) + w_j g_j = 0, \qquad 2n_j^t(u_j^t - \hat{\mathbb{E}}_{t,j}) - w_j g_j = 0,$$

where $g_j$ is any subgradient of the norm at $u_j^s - u_j^t$ (a unit vector when the difference is nonzero). Eliminating $g_j$ gives

$$u_j^s = \hat{\mathbb{E}}_{s,j} - \frac{w_j}{2n_j^s} g_j, \qquad u_j^t = \hat{\mathbb{E}}_{t,j} + \frac{w_j}{2n_j^t} g_j.$$

When the quadratic penalty terms are minimized (forcing the auxiliary variables to their weighted empirical means), the correction terms vanish and

$$u_j^s \to \hat{\mathbb{E}}_{s,j}, \qquad u_j^t \to \hat{\mathbb{E}}_{t,j}.$$

Substituting these optimal auxiliary values back into Eq. 10 yields

$$\sum_{j=1}^{B} w_j \|\hat{\mathbb{E}}_{s,j} - \hat{\mathbb{E}}_{t,j}\|,$$

which is exactly Eq. 8. Hence Eq. 10 is asymptotically equivalent to Eq. 8, with an $O(w_j/n_j^d)$ gap (from the subgradient penalties $\frac{w_j}{2n_j^s} g_j$ and $\frac{w_j}{2n_j^t} g_j$) that vanishes as $n_j^s, n_j^t \to \infty$.

**Unbiasedness of the mini-batch gradient.** We will first prove that Eq. 8 produces a biased gradient estimate on mini-batches, and then prove that Eq. 10 produces an unbiased gradient estimate.

Write the differentiable ECL (Eq. 8) as

$$\hat{L}_{ecl}(\theta) = \sum_{j=1}^{B} w_j \|\hat{\mathbb{E}}_{s,j} - \hat{\mathbb{E}}_{t,j}\|.$$

For notational clarity and for an arbitrary norm $\|\cdot\|$ introduce a subgradient selection

$$g_j \in \partial \|\hat{\mathbb{E}}_{s,j} - \hat{\mathbb{E}}_{t,j}\| \quad \text{(any choice when the difference is nonzero)}.$$

Using the chain rule for a general norm we obtain the full-data gradient

$$\nabla_\theta \hat{L}_{ecl}(\theta) = \sum_{j=1}^{B} w_j \left\langle g_j, \ \nabla_\theta \hat{\mathbb{E}}_{s,j} - \nabla_\theta \hat{\mathbb{E}}_{t,j} \right\rangle, \tag{32}$$

where, for example, the full-data weighted gradient average is

$$\nabla_\theta \hat{\mathbb{E}}_{s,j} = \frac{1}{n_j^s} \sum_{i \in D_s} \omega_{i,j}^s \nabla_\theta p^{(i)}(\theta).$$

Now consider computing the same expression on a random mini-batch. Let $\hat{\mathbb{E}}_{s,j}^{\mathrm{m}}, \hat{\mathbb{E}}_{t,j}^{\mathrm{m}}$ be the per-bin weighted means computed from the current mini-batches and choose a measurable subgradient selection $g_j^{\mathrm{m}} \in \partial \|\hat{\mathbb{E}}_{s,j}^{\mathrm{m}} - \hat{\mathbb{E}}_{t,j}^{\mathrm{m}}\|$. The mini-batch gradient contribution for bin $j$ (when using Eq. 8 directly on the mini-batch) equals

$$G_j^{\mathrm{m}} = w_j \left\langle g_j^{\mathrm{m}}, \ \nabla_\theta \hat{\mathbb{E}}_{s,j}^{\mathrm{m}} - \nabla_\theta \hat{\mathbb{E}}_{t,j}^{\mathrm{m}} \right\rangle.$$

Taking expectation over the random mini-batch sampling (the indices in the sums) and using linearity gives

$$\mathbb{E}[G_j^{\mathrm{m}}] = w_j \left( \mathbb{E}[g_j^{\mathrm{m}}]^\top \, \mathbb{E}[\nabla_\theta \hat{\mathbb{E}}_{s,j}^{\mathrm{m}} - \nabla_\theta \hat{\mathbb{E}}_{t,j}^{\mathrm{m}}] \, + \, \mathrm{Cov}\big(g_j^{\mathrm{m}}, \, \nabla_\theta \hat{\mathbb{E}}_{s,j}^{\mathrm{m}} - \nabla_\theta \hat{\mathbb{E}}_{t,j}^{\mathrm{m}}\big) \right), \tag{33}$$

where the covariance denotes the cross-covariance between the components of the subgradient vector $g_j^{\mathrm{m}}$ and the gradient estimator. The covariance need not vanish because $g_j^{\mathrm{m}}$ is a nonlinear (sub)differential selection of the same mini-batch samples that produce the per-sample gradients; hence in general

$$\mathbb{E}[G_j^{\mathrm{m}}] \neq w_j g_j^\top \big(\nabla_\theta \hat{\mathbb{E}}_{s,j} - \nabla_\theta \hat{\mathbb{E}}_{t,j}\big).$$

This equality would hold only if $g_j^{\mathrm{m}}$ were (in expectation) equal to $g_j$ and uncorrelated with the mini-batch gradient estimator — a condition that generally fails because of the nonlinear subgradient selection.

Eq. 10 (Eq.13) remedies this issue by introducing auxiliary variables $u_j^s, u_j^t$. Concretely, let us set the auxiliaries to the full-data weighted means (functions of $\theta$ but independent of the current mini-batch indices):

$$u_j^{s,\mathrm{full}} := \hat{\mathbb{E}}_{s,j} = \frac{1}{n_j^s} \sum_{i \in D_s} \omega_{i,j}^s p^{(i)}(\theta), \qquad u_j^{t,\mathrm{full}} := \hat{\mathbb{E}}_{t,j} = \frac{1}{n_j^t} \sum_{i \in D_t} \omega_{i,j}^t p^{(i)}(\theta).$$

Define the fixed unit vector

$$v_j^{\mathrm{full}} := \frac{u_j^{s,\mathrm{full}} - u_j^{t,\mathrm{full}}}{\|u_j^{s,\mathrm{full}} - u_j^{t,\mathrm{full}}\|}.$$

If we compute the mini-batch gradient of Eq. 10 while treating $u_j^d = u_j^{d,\mathrm{full}}$ as fixed (i.e. independent of the current mini-batch samples), the bin-$j$ contribution equals

$$\tilde{G}_j^{\mathrm{m}} = w_j \left\langle v_j^{\mathrm{full}}, \, \frac{1}{|D_s^{\mathrm{m}}|} \sum_{i \in D_s^{\mathrm{m}}} \omega_{i,j}^s \nabla_\theta p^{(i)}(\theta) - \frac{1}{|D_t^{\mathrm{m}}|} \sum_{i \in D_t^{\mathrm{m}}} \omega_{i,j}^t \nabla_\theta p^{(i)}(\theta) \right\rangle.$$

Taking expectation over the random mini-batch sampling (the indices in the sums) and using linearity gives

$$\mathbb{E}[\tilde{G}_j^{\mathrm{m}}] = w_j \left\langle v_j^{\mathrm{full}}, \, \frac{1}{n_j^s} \sum_{i \in D_s} \omega_{i,j}^s \nabla_\theta p^{(i)}(\theta) - \frac{1}{n_j^t} \sum_{i \in D_t} \omega_{i,j}^t \nabla_\theta p^{(i)}(\theta) \right\rangle.$$

The right-hand side is exactly the full-data bin-$j$ term in Eq. 32; summing over $j$ yields

$$\mathbb{E}\left[ \sum_{j=1}^B \tilde{G}_j^{\mathrm{m}} \right] = \nabla_\theta \hat{L}_{ecl}(\theta).$$

Thus, when Eq. 10 is used with auxiliaries taken from an estimate independent of the current mini-batch (e.g. full-data means, a large buffer, or a slow running average), the mini-batch gradient is an unbiased estimator of the full-sample gradient.

**Algorithm 2** Top-label ECL Mini-Batch.

1: **Input:**
2:    bins $j = 1 \ldots B$, hyperparameters $\lambda, \alpha_{\text{ema}}, N_{\text{prox}}$;
3:    $u_j^s = 0 \in \mathbb{R}, \forall j$; $u_j^t = 0 \in \mathbb{R}, \forall j$;
4: **for** each iteration **do**
5:    Sample mini-batches $D_s^m, D_t^m$;
6:    Compute weights $\omega_{ij}^s, \omega_{ij}^t$;
7:    $n_{s,j} \leftarrow \sum_{i \in D_s^m} \omega_{ij}^s$; $n_{t,j} \leftarrow \sum_{i \in D_t^m} \omega_{ij}^t$;
8:    $m_{s,j} \leftarrow \sum_{i \in D_s^m} \omega_{ij}^s P(Y^* = \hat{Y} | X = x_i)$;
9:    $m_{t,j} \leftarrow \sum_{i \in D_t^m} \omega_{ij}^t P(Y^* = \hat{Y} | X = x_i)$;
10:    $w_j \leftarrow n_{t,j} / \sum_{r=1}^{B} n_{t,r}$;
11:    $L_{\text{ecl}} \leftarrow 0$;
12:    **for** each bin $j$ **do**
13:       $u_s, u_t \leftarrow$ cached $u_j^s, u_j^t$
14:       **for** $i = 1$ to $N_{\text{prox}}$ **do**
15:          $v_s \leftarrow (m_{s,j}/n_{s,j}) - u_t, \tau_s = \dfrac{w_j}{2n_{s,j}}$
16:          $u_s \leftarrow u_t + \text{shrink}(v_s, \tau_s)$
17:          $v_t \leftarrow (m_{t,j}/n_{t,j}) - u_s, \tau_t = \dfrac{w_j}{2n_{t,j}}$
18:          $u_t \leftarrow u_s + \text{shrink}(v_t, \tau_t)$
19:       **end for**
20:       $\tilde{u}_j^s, \tilde{u}_j^t \leftarrow u_s.\text{detach}(), u_t.\text{detach}()$
21:       $u_j^s \leftarrow (1 - \alpha_{\text{ema}}) u_j^s + \alpha_{\text{ema}} \tilde{u}_j^s$
22:       $u_j^t \leftarrow (1 - \alpha_{\text{ema}}) u_j^t + \alpha_{\text{ema}} \tilde{u}_j^t$
23:       $L_{\text{ecl}} += \sum_{i \in D_s^m} \omega_{ij}^s \| \tilde{u}_j^s - P(Y^* = \hat{Y} | X = x_i) \|^2$
24:       $L_{\text{ecl}} += \sum_{i \in D_t^m} \omega_{ij}^t \| \tilde{u}_j^t - P(Y^* = \hat{Y} | X = x_i) \|^2$
25:    **end for**
26:    Compute the cross-entropy loss $L_{\text{ce}}$
27:    Backpropagate $L_{\text{ce}} + \lambda L_{\text{ecl}}$ and update $\theta$
28: **end for**
29: **Return:** $\theta$

**Algorithm 3** Class-wise ECL Mini-Batch.

1: **Input:**
2:    bins $j = 1 \ldots B$, hyperparameters $\lambda, \alpha_{\text{ema}}, N_{\text{prox}}$;
3:    $u_{k,j}^s = 0 \in \mathbb{R}, \forall k, j$; $u_{k,j}^t = 0 \in \mathbb{R}, \forall k, j$;
4: **for** each iteration **do**
5:    Sample mini-batches $D_s^m, D_t^m$; $L_{\text{ecl}} \leftarrow 0$;
6:    **for** each class $k = 1$ to $K$ **do**
7:       Compute weights $\omega_{k,ij}^s, \omega_{k,ij}^t$;
8:       $n_{s,j} \leftarrow \sum_{i \in D_s^m} \omega_{k,ij}^s$; $n_{t,j} \leftarrow \sum_{i \in D_t^m} \omega_{k,ij}^t$;
9:       $m_{s,j} \leftarrow \sum_{i \in D_s^m} \omega_{k,ij}^s p_k^{(i)}(\theta)$;
10:       $m_{t,j} \leftarrow \sum_{i \in D_t^m} \omega_{k,ij}^t p_k^{(i)}(\theta)$;
11:       $w_{k,j} \leftarrow n_{t,j} / \sum_{r=1}^{B} n_{t,r}$;
12:       **for** each bin $j$ **do**
13:          $u_s, u_t \leftarrow$ cached $u_{k,j}^s, u_{k,j}^t$
14:          **for** $i = 1$ to $N_{\text{prox}}$ **do**
15:             $v_s \leftarrow (m_{s,j}/n_{s,j}) - u_t, \tau_s = \dfrac{w_{k,j}}{2n_{s,j}}$
16:             $u_s \leftarrow u_t + \text{shrink}(v_s, \tau_s)$
17:             $v_t \leftarrow (m_{t,j}/n_{t,j}) - u_s, \tau_t = \dfrac{w_{k,j}}{2n_{t,j}}$
18:             $u_t \leftarrow u_s + \text{shrink}(v_t, \tau_t)$
19:          **end for**
20:          $\tilde{u}_{k,j}^s, \tilde{u}_{k,j}^t \leftarrow u_s.\text{detach}(), u_t.\text{detach}()$
21:          $u_{k,j}^s \leftarrow (1 - \alpha_{\text{ema}}) u_{k,j}^s + \alpha_{\text{ema}} \tilde{u}_{k,j}^s$
22:          $u_{k,j}^t \leftarrow (1 - \alpha_{\text{ema}}) u_{k,j}^t + \alpha_{\text{ema}} \tilde{u}_{k,j}^t$
23:          $L_{\text{ecl}} += \sum_{i \in D_s^m} \omega_{k,ij}^s \| \tilde{u}_{k,j}^s - p_k^{(i)}(\theta) \|^2$
24:          $L_{\text{ecl}} += \sum_{i \in D_t^m} \omega_{k,ij}^t \| \tilde{u}_{k,j}^t - p_k^{(i)}(\theta) \|^2$
25:       **end for**
26:    **end for**
27:    Compute the cross-entropy loss $L_{\text{ce}}$
28:    Backpropagate $L_{\text{ce}} + \lambda L_{\text{ecl}}$ and update $\theta$
29: **end for**
30: **Return:** $\theta$

## I. Extension of ECL Mini-Batch Training

Algorithm 1 details the ECL mini-batch training for canonical calibration. Here, we present the analogous algorithms for top-label calibration (Algorithm 2) and class-wise calibration (Algorithm 3). They employ the same auxiliary variable strategy to resolve the bias in mini-batch gradients. In Algorithm 2, $P(Y^* = \hat{Y} | X = x)$ can be obtained by training a binary classifier where the label is $1_{Y^* = \hat{Y}}$ and the input data is $X$. Moreover, this binary classifier does not need to be trained separately. It can be added to the original classifier as a classification head and trained end-to-end with the original classifier (freeze the backbone when training this classification head).

## J. Results

**Other experimental settings:** The batch size in the experiment is uniformly set to 100. Adam optimizer with a learning rate of 0.001 is used to train the classifier for 100 epochs. All experiments were conducted on Intel® Core$^{TM}$ I7-10700 CPU with 3.70GHz and 125.5GB memory, 10 NVIDIA GeForce RTX 3090 graphics cards (each with 24GB of video memory), Ubuntu 20.04.3 LTS, Python 3.11.11, and Torch 2.4.1+cu118. We calibrate the classification head used to estimate $P(Y|X)$

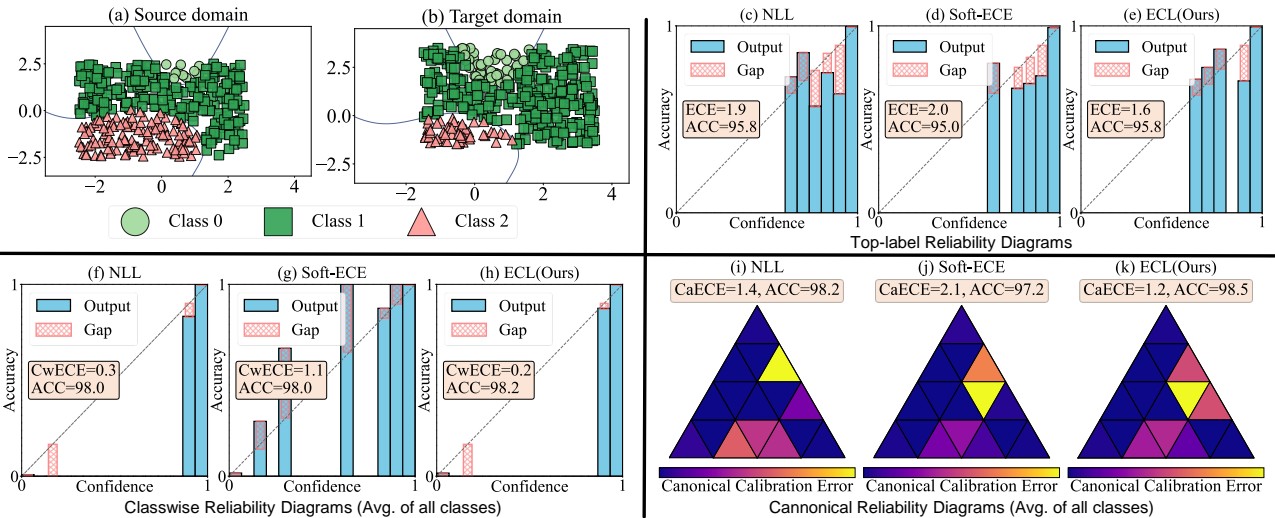

*Figure 3.* The calibration results are presented using simulated data under a uniformly distributed covariate shift. From the calibration metric on the target domain and the reliability diagram of the calibrated classifier, ECL achieves the smallest calibration error.

(or $P(Y^* = \hat{Y}|X)$ for top-label calibration) on the source domain using Soft-ECE loss. This classification head has the same network structure as the classification head in the original classifier, and uses the same hyperparameters during training. All images in the digit recognition dataset were standardized to 3-channel RGB format and resized to a resolution of $28\times28$ pixels. All images in PACS and ImageNet-Sketch were standardized to 3-channel RGB format and resized to a resolution of $224\times224$ pixels.

### J.1. Results on Simulated Covariate Shifts Data

Figure 3 shows the calibration results under a uniformly distributed covariate shift, complementing the normally distributed case in Figure 2. Consistent with the normal case, ECL achieves the lowest calibration error across all three paradigms.

### J.2. Results for Top-label Calibration

Table 4 details the top-label calibration performance on the PACS and ImageNet-Sketch datasets. Several key observations can be drawn regarding the effectiveness of ECL. First, regarding robustness to large shifts, on the ImageNet-Sketch dataset—which presents a severe distribution shift (source ImageNet vs. target Sketch)—uncalibrated models exhibit extreme ECE values exceeding 55%. ECL substantially reduces these errors (often to below 15% across the tested architectures), demonstrating its capability to handle substantial domain gaps. Second, while PseudoCal serves as a strong baseline, ECL is generally competitive and frequently achieves a lower ECE. For instance, in the PACS ($\to$ Cartoon) task using Wide-Res50, ECL achieves an ECE of 7.61%, outperforming PseudoCal (16.24%) and improving upon DRL (8.36%). Finally, the method remains effective across diverse architectures, from standard CNNs (ResNet, DenseNet) to Vision Transformers (ViT-L), suggesting that the Expectation Consistency condition captures a model-agnostic principle.

### J.3. Results for Class-wise Calibration

Table 5 reports the Class-wise ECE (CwECE) results. Two major trends are evident from the experimental data. First, unlike top-label calibration which focuses on the predicted class only, class-wise calibration requires precision across all categories. ECL achieves the lowest (or near-lowest) CwECE in many experimental settings (spanning datasets and models), indicating that it improves calibration not only for the dominant class. Second, regarding handling hard tasks, on the Digit recognition benchmarks (included in Table 5), the advantage of ECL is most prominent on the SVHN dataset. For the LeNet-5 architecture, ECL reduces CwECE from 15.8% (Uncal) to 5.88%, improving upon most baselines (e.g., PseudoCal at 12.7%). This suggests that ECL's auxiliary variable optimization can be particularly effective in scenarios with complex background noise and lower image quality.

*Table 4.* ECE (%) for top-label calibration on PACS and ImageNet-Sketch datasets. The reported results represent the mean and standard deviation derived from ten runs.

| Datasets | Uncal | Soft-ECE | DECE | KDE | ECE↓ TS | TransCal | DRL | PseudoCal | ECL (Ours) | Oracle↓ | ΔACC(%) |
|---|---|---|---|---|---|---|---|---|---|---|---|
| **→ Photo** | | | | | | | | | | | |
| ResNet50 | $22.3_{\pm2.16}$ | $22.1_{\pm1.83}$ | $22.8_{\pm1.99}$ | $21.8_{\pm1.72}$ | $20.9_{\pm1.53}$ | $22.2_{\pm1.66}$ | $9.02_{\pm0.57}$ | $7.33_{\pm0.44}$ | $6.87_{\pm0.34}$ | $3.84_{\pm0.23}$ | $+0.72_{\pm0.17}$ |
| DenseNet121 | $9.78_{\pm0.96}$ | $9.88_{\pm0.91}$ | $9.54_{\pm0.91}$ | $10.2_{\pm0.86}$ | $9.63_{\pm0.69}$ | $9.31_{\pm0.63}$ | $7.91_{\pm0.41}$ | $6.61_{\pm0.54}$ | $5.96_{\pm0.27}$ | $1.84_{\pm0.13}$ | $-0.83_{\pm0.23}$ |
| Wide-Res50 | $16.9_{\pm1.42}$ | $17.2_{\pm1.24}$ | $17.8_{\pm1.39}$ | $16.8_{\pm1.18}$ | $16.2_{\pm1.37}$ | $7.27_{\pm0.47}$ | $4.39_{\pm0.42}$ | $2.83_{\pm0.11}$ | $2.68_{\pm0.33}$ | $1.59_{\pm0.17}$ | $+0.69_{\pm0.22}$ |
| **→ Art** | | | | | | | | | | | |
| ResNet50 | $33.1_{\pm3.24}$ | $32.1_{\pm2.97}$ | $33.2_{\pm3.11}$ | $31.6_{\pm3.06}$ | $31.9_{\pm3.24}$ | $17.1_{\pm1.51}$ | $17.1_{\pm1.14}$ | $7.88_{\pm0.78}$ | $7.22_{\pm0.53}$ | $2.12_{\pm0.08}$ | $-1.24_{\pm0.41}$ |
| DenseNet121 | $23.2_{\pm2.04}$ | $22.8_{\pm1.94}$ | $23.6_{\pm2.14}$ | $23.1_{\pm1.86}$ | $22.7_{\pm1.96}$ | $22.4_{\pm1.88}$ | $6.16_{\pm0.59}$ | $9.94_{\pm0.74}$ | $5.89_{\pm0.36}$ | $2.24_{\pm0.21}$ | $+1.06_{\pm0.39}$ |
| Wide-Res50 | $29.9_{\pm2.78}$ | $29.3_{\pm2.57}$ | $30.4_{\pm2.64}$ | $29.7_{\pm2.42}$ | $30.1_{\pm2.53}$ | $16.1_{\pm1.23}$ | $15.8_{\pm1.42}$ | $8.43_{\pm0.67}$ | $7.97_{\pm0.52}$ | $3.14_{\pm0.24}$ | $-0.96_{\pm0.29}$ |
| **→ Cartoon** | | | | | | | | | | | |
| ResNet50 | $25.1_{\pm2.26}$ | $25.1_{\pm2.08}$ | $25.3_{\pm2.42}$ | $24.9_{\pm2.07}$ | $24.8_{\pm1.91}$ | $25.2_{\pm2.39}$ | $6.69_{\pm0.48}$ | $5.71_{\pm0.43}$ | $5.46_{\pm0.43}$ | $2.73_{\pm0.26}$ | $+0.56_{\pm0.12}$ |
| DenseNet121 | $18.4_{\pm1.48}$ | $18.7_{\pm1.36}$ | $17.8_{\pm1.56}$ | $18.4_{\pm1.44}$ | $18.3_{\pm1.73}$ | $11.3_{\pm0.91}$ | $10.9_{\pm1.21}$ | $2.21_{\pm0.09}$ | $2.04_{\pm0.16}$ | $2.04_{\pm0.18}$ | $-0.74_{\pm0.22}$ |
| Wide-Res50 | $25.4_{\pm1.98}$ | $24.9_{\pm1.88}$ | $25.9_{\pm2.06}$ | $25.6_{\pm1.72}$ | $25.2_{\pm1.76}$ | $23.7_{\pm1.88}$ | $8.36_{\pm0.67}$ | $16.24_{\pm1.44}$ | $7.61_{\pm0.28}$ | $2.73_{\pm0.19}$ | $+1.26_{\pm0.39}$ |
| **→ Sketch** | | | | | | | | | | | |
| ResNet50 | $23.1_{\pm1.87}$ | $23.9_{\pm1.97}$ | $22.9_{\pm2.06}$ | $23.4_{\pm1.88}$ | $23.4_{\pm2.24}$ | $11.4_{\pm0.96}$ | $16.2_{\pm1.29}$ | $10.9_{\pm0.98}$ | $10.3_{\pm0.82}$ | $1.54_{\pm0.13}$ | $-1.53_{\pm0.47}$ |
| DenseNet121 | $23.6_{\pm1.57}$ | $22.8_{\pm1.43}$ | $23.8_{\pm1.72}$ | $23.2_{\pm1.54}$ | $22.9_{\pm1.96}$ | $9.09_{\pm0.79}$ | $3.39_{\pm0.19}$ | $5.39_{\pm0.51}$ | $3.17_{\pm0.28}$ | $2.66_{\pm0.16}$ | $+0.86_{\pm0.21}$ |
| Wide-Res50 | $19.2_{\pm1.41}$ | $19.6_{\pm1.36}$ | $18.8_{\pm1.51}$ | $19.1_{\pm1.26}$ | $18.9_{\pm1.66}$ | $10.01_{\pm0.96}$ | $6.81_{\pm0.68}$ | $2.79_{\pm0.21}$ | $2.67_{\pm0.22}$ | $2.69_{\pm0.28}$ | $-0.48_{\pm0.13}$ |
| **→ Sketch** (I-S) | | | | | | | | | | | |
| ResNet152 | $64.3_{\pm4.48}$ | $63.6_{\pm4.12}$ | $65.1_{\pm4.64}$ | $63.4_{\pm4.36}$ | $62.8_{\pm4.19}$ | $60.1_{\pm3.94}$ | $33.3_{\pm2.34}$ | $17.3_{\pm1.68}$ | $14.6_{\pm0.58}$ | $1.54_{\pm0.09}$ | $+0.92_{\pm0.31}$ |
| DenseNet161 | $69.1_{\pm3.62}$ | $68.7_{\pm3.47}$ | $69.8_{\pm3.86}$ | $68.3_{\pm3.57}$ | $68.3_{\pm4.66}$ | $58.4_{\pm4.33}$ | $36.9_{\pm2.69}$ | $13.2_{\pm1.21}$ | $11.7_{\pm0.46}$ | $1.27_{\pm0.14}$ | $+1.39_{\pm0.59}$ |
| ViT-L | $55.8_{\pm4.34}$ | $55.1_{\pm4.07}$ | $56.7_{\pm4.49}$ | $54.9_{\pm4.26}$ | $53.7_{\pm4.16}$ | $32.7_{\pm2.38}$ | $27.1_{\pm1.79}$ | $15.7_{\pm1.24}$ | $12.9_{\pm0.54}$ | $1.47_{\pm0.11}$ | $+0.92_{\pm0.28}$ |

## J.4. Results for Canonical Calibration

The results for Canonical Calibration, measured by $\text{ECE}^{KDE}$ in Table 6, further confirm the comprehensive efficacy of ECL. The findings highlight two main points. First, canonical calibration is the most rigorous standard as it requires the entire probability vector to be calibrated. The differentiable baseline KDE loss operates within-domain and does not explicitly address the covariate shift, often performing similarly to the uncalibrated baseline in our setting (e.g., Table 6 ResNet50 on Photo). In contrast, ECL explicitly minimizes the cross-domain discrepancy of probability expectations, frequently achieving the best (or near-best) $\text{ECE}^{KDE}$ scores. Second, in terms of accuracy, the reduction in $\text{ECE}^{KDE}$ is often achieved with limited impact on classification accuracy; while ΔACC is positive in many cases, slight accuracy drops can still occur for some architectures/tasks.

## J.5. Ablation Experiments

**Mini-Batch Non-Trainable ECL vs. Mini-Batch Trainable ECL:** To understand the efficacy of our proposed mini-batch training strategy involving auxiliary variables, we compare our full method (*Mini-Batch Trainable ECL*) against a baseline variant *Mini-Batch Non-Trainable ECL* (it refers to directly calculating the differentiable ECL loss (Eq. 8) on mini-batch data). Table 7 presents the comparison results on the Digit (→ MNIST) and PACS (→ Photo) tasks. Overall, *Mini-Batch Trainable ECL* tends to be more stable and achieves better calibration in most cases, while *Mini-Batch Non-Trainable ECL* can occasionally be competitive on some metrics/architectures. This supports that, beyond the objective itself, the bias-corrected optimization strategy is important for reliably realizing ECL's benefits. Regarding classification accuracy (ΔACC), both variants largely maintain or improve performance, with *Mini-Batch Trainable ECL* showing more consistent gains in our reported experiments.

**Loss Weight:** To maintain the equal importance of $L_{ce}$ and $L_{ecl}$, we set the regularization weight as $\lambda = \beta^\gamma$. Here, $\beta = \left(\sum_i \mathcal{L}_{ce}^{(i)}\right) / \left(\sum_i \mathcal{L}_{ecl}^{(i)}\right)$ acts as a baseline balancing factor between the cross-entropy loss and the calibration loss, where $i$ represents the $i$-th iteration. The exponent $\gamma$ serves as a non-linear scaling factor to adjust the sensitivity of the regularization: a higher $\gamma$ (when $\beta > 1$) or lower $\gamma$ (when $\beta < 1$) intensifies the dominance of the calibration term. We investigate the impact of $\gamma$ by experimenting with values ranging from 0.5 to 1.5, and Table 8 suggests that $\gamma = 1.0$ is a reasonable default choice in our evaluated settings (Digit → MNIST and PACS → Photo).

*Table 5.* CwECE (%) for class-wise calibration on Digit, PACS and ImageNet-Sketch datasets. The reported results represent the mean and standard deviation derived from ten runs.

| Datasets | Uncal | Soft-ECE | DECE | KDE | TS | TransCal | DRL | PseudoCal | ECL (Ours) | Oracle ↓ | ΔACC(%) |
|---|---|---|---|---|---|---|---|---|---|---|---|
| | | | | | **CwECE ↓** | | | | | | |
| **Digit** | | | | | | | | | | | |
| → *MNIST* | | | | | | | | | | | |
| LeNet-5 | $5.41_{\pm0.47}$ | $5.54_{\pm0.37}$ | $5.31_{\pm0.43}$ | $5.49_{\pm0.39}$ | $5.18_{\pm0.33}$ | $4.92_{\pm0.39}$ | $3.79_{\pm0.24}$ | $1.86_{\pm0.12}$ | $1.66_{\pm0.12}$ | $0.16_{\pm0.01}$ | $-0.44_{\pm0.09}$ |
| ResNet20 | $3.14_{\pm0.31}$ | $3.23_{\pm0.22}$ | $3.13_{\pm0.28}$ | $3.21_{\pm0.22}$ | $3.06_{\pm0.23}$ | $2.47_{\pm0.17}$ | $1.94_{\pm0.14}$ | $1.46_{\pm0.14}$ | $1.41_{\pm0.11}$ | $0.39_{\pm0.01}$ | $+0.62_{\pm0.11}$ |
| DenseNet40 | $4.69_{\pm0.41}$ | $4.74_{\pm0.37}$ | $4.51_{\pm0.29}$ | $4.66_{\pm0.43}$ | $4.46_{\pm0.39}$ | $3.94_{\pm0.27}$ | $2.81_{\pm0.21}$ | $1.77_{\pm0.19}$ | $1.57_{\pm0.12}$ | $0.38_{\pm0.06}$ | $+0.23_{\pm0.11}$ |
| → *USPS* | | | | | | | | | | | |
| LeNet-5 | $6.87_{\pm0.54}$ | $6.96_{\pm0.47}$ | $6.77_{\pm0.57}$ | $6.91_{\pm0.44}$ | $6.63_{\pm0.46}$ | $5.84_{\pm0.36}$ | $4.13_{\pm0.31}$ | $2.19_{\pm0.19}$ | $2.11_{\pm0.14}$ | $0.57_{\pm0.03}$ | $-0.33_{\pm0.09}$ |
| ResNet20 | $2.54_{\pm0.23}$ | $2.66_{\pm0.19}$ | $2.46_{\pm0.24}$ | $2.59_{\pm0.23}$ | $2.46_{\pm0.22}$ | $2.14_{\pm0.16}$ | $1.73_{\pm0.14}$ | $1.17_{\pm0.07}$ | $1.24_{\pm0.09}$ | $0.63_{\pm0.01}$ | $+0.42_{\pm0.18}$ |
| DenseNet40 | $3.99_{\pm0.36}$ | $4.03_{\pm0.29}$ | $3.83_{\pm0.33}$ | $3.99_{\pm0.28}$ | $3.63_{\pm0.24}$ | $3.27_{\pm0.19}$ | $2.14_{\pm0.14}$ | $1.48_{\pm0.12}$ | $1.18_{\pm0.12}$ | $0.72_{\pm0.06}$ | $-0.16_{\pm0.04}$ |
| → *SVHN* | | | | | | | | | | | |
| LeNet-5 | $15.8_{\pm1.26}$ | $15.8_{\pm1.17}$ | $15.4_{\pm1.37}$ | $16.2_{\pm1.26}$ | $15.4_{\pm1.19}$ | $14.4_{\pm1.12}$ | $8.54_{\pm0.61}$ | $12.7_{\pm0.92}$ | $5.88_{\pm0.43}$ | $0.44_{\pm0.02}$ | $+0.84_{\pm0.24}$ |
| ResNet20 | $18.4_{\pm1.44}$ | $18.2_{\pm1.34}$ | $18.8_{\pm1.54}$ | $18.1_{\pm1.29}$ | $18.2_{\pm1.37}$ | $15.1_{\pm1.17}$ | $9.86_{\pm0.74}$ | $11.2_{\pm0.88}$ | $8.97_{\pm0.59}$ | $0.22_{\pm0.01}$ | $-1.04_{\pm0.36}$ |
| DenseNet40 | $21.3_{\pm1.64}$ | $21.8_{\pm1.53}$ | $21.2_{\pm1.76}$ | $21.4_{\pm1.49}$ | $20.6_{\pm1.49}$ | $18.4_{\pm1.31}$ | $11.3_{\pm0.84}$ | $15.2_{\pm1.14}$ | $8.16_{\pm0.57}$ | $0.39_{\pm0.03}$ | $+0.53_{\pm0.17}$ |
| **PACS** | | | | | | | | | | | |
| → *Photo* | | | | | | | | | | | |
| ResNet50 | $7.87_{\pm0.31}$ | $7.89_{\pm0.43}$ | $7.77_{\pm0.38}$ | $7.84_{\pm0.44}$ | $7.62_{\pm0.32}$ | $6.86_{\pm0.31}$ | $5.99_{\pm0.26}$ | $3.24_{\pm0.21}$ | $2.92_{\pm0.12}$ | $0.58_{\pm0.01}$ | $+0.48_{\pm0.09}$ |
| DenseNet121 | $8.53_{\pm0.47}$ | $8.64_{\pm0.44}$ | $8.46_{\pm0.52}$ | $8.59_{\pm0.42}$ | $8.37_{\pm0.37}$ | $7.58_{\pm0.29}$ | $6.28_{\pm0.23}$ | $3.87_{\pm0.24}$ | $3.56_{\pm0.19}$ | $0.61_{\pm0.01}$ | $+0.29_{\pm0.11}$ |
| Wide-Res50 | $6.99_{\pm0.38}$ | $6.99_{\pm0.32}$ | $6.81_{\pm0.39}$ | $6.99_{\pm0.32}$ | $6.78_{\pm0.32}$ | $6.17_{\pm0.31}$ | $5.24_{\pm0.26}$ | $2.83_{\pm0.14}$ | $2.58_{\pm0.09}$ | $0.48_{\pm0.04}$ | $+0.34_{\pm0.12}$ |
| → *Art* | | | | | | | | | | | |
| ResNet50 | $13.3_{\pm0.64}$ | $13.9_{\pm0.73}$ | $13.2_{\pm0.78}$ | $13.3_{\pm0.64}$ | $12.8_{\pm0.54}$ | $11.1_{\pm0.47}$ | $8.58_{\pm0.36}$ | $5.28_{\pm0.23}$ | $4.86_{\pm0.24}$ | $0.84_{\pm0.02}$ | $-0.28_{\pm0.12}$ |
| DenseNet121 | $14.4_{\pm0.73}$ | $14.7_{\pm0.82}$ | $13.6_{\pm0.84}$ | $14.3_{\pm0.77}$ | $13.1_{\pm0.63}$ | $11.4_{\pm0.51}$ | $9.28_{\pm0.47}$ | $5.89_{\pm0.39}$ | $5.94_{\pm0.31}$ | $0.94_{\pm0.04}$ | $-0.18_{\pm0.11}$ |
| Wide-Res50 | $12.6_{\pm0.56}$ | $12.8_{\pm0.67}$ | $12.7_{\pm0.71}$ | $12.8_{\pm0.63}$ | $11.8_{\pm0.56}$ | $9.96_{\pm0.46}$ | $7.84_{\pm0.32}$ | $4.58_{\pm0.23}$ | $4.13_{\pm0.17}$ | $0.78_{\pm0.08}$ | $+0.44_{\pm0.14}$ |
| → *Cartoon* | | | | | | | | | | | |
| ResNet50 | $16.4_{\pm0.84}$ | $16.6_{\pm0.88}$ | $16.4_{\pm0.93}$ | $16.3_{\pm0.92}$ | $15.4_{\pm0.73}$ | $14.1_{\pm0.64}$ | $10.3_{\pm0.54}$ | $6.86_{\pm0.43}$ | $6.46_{\pm0.34}$ | $1.17_{\pm0.08}$ | $+0.63_{\pm0.24}$ |
| DenseNet121 | $17.1_{\pm0.98}$ | $17.3_{\pm1.08}$ | $17.1_{\pm1.16}$ | $17.3_{\pm1.02}$ | $16.6_{\pm0.87}$ | $14.7_{\pm0.73}$ | $10.9_{\pm0.67}$ | $7.53_{\pm0.56}$ | $7.13_{\pm0.49}$ | $1.27_{\pm0.09}$ | $+0.32_{\pm0.19}$ |
| Wide-Res50 | $15.6_{\pm0.81}$ | $15.9_{\pm0.81}$ | $15.3_{\pm0.84}$ | $15.9_{\pm0.81}$ | $14.7_{\pm0.72}$ | $12.9_{\pm0.59}$ | $9.86_{\pm0.51}$ | $6.16_{\pm0.41}$ | $5.83_{\pm0.31}$ | $1.06_{\pm0.03}$ | $+0.51_{\pm0.14}$ |
| → *Sketch* | | | | | | | | | | | |
| ResNet50 | $19.6_{\pm1.18}$ | $19.7_{\pm1.23}$ | $19.3_{\pm1.32}$ | $19.6_{\pm1.21}$ | $18.6_{\pm1.01}$ | $16.2_{\pm0.96}$ | $13.4_{\pm0.86}$ | $8.82_{\pm0.72}$ | $9.28_{\pm0.67}$ | $1.43_{\pm0.11}$ | $-0.88_{\pm0.21}$ |
| DenseNet121 | $20.3_{\pm1.24}$ | $20.4_{\pm1.37}$ | $19.9_{\pm1.41}$ | $20.1_{\pm1.26}$ | $19.2_{\pm1.14}$ | $17.7_{\pm1.03}$ | $13.9_{\pm0.92}$ | $9.53_{\pm0.88}$ | $8.91_{\pm0.74}$ | $1.53_{\pm0.14}$ | $+0.28_{\pm0.19}$ |
| Wide-Res50 | $18.8_{\pm1.06}$ | $18.8_{\pm1.18}$ | $18.6_{\pm1.27}$ | $18.9_{\pm1.06}$ | $17.9_{\pm0.97}$ | $15.4_{\pm0.84}$ | $12.9_{\pm0.72}$ | $8.16_{\pm0.64}$ | $7.87_{\pm0.54}$ | $1.36_{\pm0.07}$ | $+0.47_{\pm0.26}$ |
| **I-S** | | | | | | | | | | | |
| → *Sketch* | | | | | | | | | | | |
| ResNet152 | $22.6_{\pm1.36}$ | $22.6_{\pm1.43}$ | $21.9_{\pm1.56}$ | $22.6_{\pm1.37}$ | $21.3_{\pm1.28}$ | $18.6_{\pm1.13}$ | $14.2_{\pm0.97}$ | $10.3_{\pm0.84}$ | $9.86_{\pm0.73}$ | $1.64_{\pm0.12}$ | $+0.84_{\pm0.37}$ |
| DenseNet161 | $23.4_{\pm1.43}$ | $23.3_{\pm1.59}$ | $22.6_{\pm1.62}$ | $23.6_{\pm1.44}$ | $22.2_{\pm1.34}$ | $19.4_{\pm1.22}$ | $15.1_{\pm1.07}$ | $11.3_{\pm0.94}$ | $10.7_{\pm0.82}$ | $1.76_{\pm0.14}$ | $-0.69_{\pm0.27}$ |
| ViT-L | $12.7_{\pm0.87}$ | $12.9_{\pm0.96}$ | $12.4_{\pm0.96}$ | $12.9_{\pm0.84}$ | $11.9_{\pm0.73}$ | $10.4_{\pm0.66}$ | $7.83_{\pm0.54}$ | $5.54_{\pm0.44}$ | $5.18_{\pm0.31}$ | $0.93_{\pm0.09}$ | $+1.26_{\pm0.26}$ |

*Table 6.* $\text{ECE}^{KDE}$ (%) for canonical calibration on Digit, PACS, and ImageNet-Sketch datasets. The reported results represent the mean and standard deviation derived from ten runs.

| Datasets | Uncal | Soft-ECE | DECE | KDE | $\text{ECE}^{KDE}\downarrow$
TS | TransCal | DRL | PseudoCal | ECL (Ours) | Oracle $\downarrow$ | $\triangle\text{ACC}(\%)$ |
|---|---|---|---|---|---|---|---|---|---|---|---|
| **Digit** | | | | | | | | | | | |
| $\to$ *MNIST* | | | | | | | | | | | |
| LeNet-5 | $5.16_{\pm0.39}$ | $5.19_{\pm0.31}$ | $5.07_{\pm0.38}$ | $5.19_{\pm0.26}$ | $4.92_{\pm0.22}$ | $4.68_{\pm0.31}$ | $3.52_{\pm0.22}$ | $1.77_{\pm0.13}$ | $1.58_{\pm0.09}$ | $0.21_{\pm0.02}$ | $-0.32_{\pm0.08}$ |
| ResNet20 | $2.97_{\pm0.23}$ | $3.07_{\pm0.18}$ | $2.84_{\pm0.26}$ | $3.01_{\pm0.13}$ | $2.73_{\pm0.17}$ | $2.29_{\pm0.14}$ | $1.72_{\pm0.13}$ | $1.36_{\pm0.11}$ | $1.29_{\pm0.04}$ | $0.39_{\pm0.02}$ | $+0.54_{\pm0.12}$ |
| DenseNet40 | $4.37_{\pm0.34}$ | $4.49_{\pm0.26}$ | $4.24_{\pm0.39}$ | $4.34_{\pm0.26}$ | $4.17_{\pm0.28}$ | $3.67_{\pm0.24}$ | $2.68_{\pm0.16}$ | $1.61_{\pm0.12}$ | $1.42_{\pm0.13}$ | $0.32_{\pm0.04}$ | $+0.19_{\pm0.02}$ |
| $\to$ *USPS* | | | | | | | | | | | |
| LeNet-5 | $6.43_{\pm0.42}$ | $6.54_{\pm0.39}$ | $6.34_{\pm0.38}$ | $6.48_{\pm0.39}$ | $6.23_{\pm0.28}$ | $5.42_{\pm0.29}$ | $3.86_{\pm0.23}$ | $2.04_{\pm0.12}$ | $1.96_{\pm0.14}$ | $0.48_{\pm0.04}$ | $-0.22_{\pm0.06}$ |
| ResNet20 | $2.38_{\pm0.22}$ | $2.42_{\pm0.16}$ | $2.24_{\pm0.24}$ | $2.42_{\pm0.09}$ | $2.16_{\pm0.14}$ | $1.97_{\pm0.11}$ | $1.54_{\pm0.14}$ | $1.12_{\pm0.07}$ | $1.17_{\pm0.04}$ | $0.51_{\pm0.06}$ | $+0.36_{\pm0.18}$ |
| DenseNet40 | $3.72_{\pm0.21}$ | $3.83_{\pm0.22}$ | $3.68_{\pm0.27}$ | $3.77_{\pm0.23}$ | $3.47_{\pm0.21}$ | $2.93_{\pm0.17}$ | $1.97_{\pm0.13}$ | $1.37_{\pm0.09}$ | $1.04_{\pm0.04}$ | $0.72_{\pm0.01}$ | $-0.11_{\pm0.01}$ |
| $\to$ *SVHN* | | | | | | | | | | | |
| LeNet-5 | $14.7_{\pm0.97}$ | $15.3_{\pm0.88}$ | $14.7_{\pm1.06}$ | $14.6_{\pm0.89}$ | $13.9_{\pm0.88}$ | $13.7_{\pm0.83}$ | $7.84_{\pm0.54}$ | $11.3_{\pm0.73}$ | $5.26_{\pm0.39}$ | $0.39_{\pm0.06}$ | $+0.69_{\pm0.26}$ |
| ResNet20 | $17.6_{\pm1.13}$ | $17.2_{\pm1.07}$ | $17.6_{\pm1.28}$ | $17.3_{\pm0.94}$ | $16.9_{\pm1.03}$ | $14.2_{\pm0.84}$ | $8.81_{\pm0.69}$ | $10.3_{\pm0.74}$ | $8.23_{\pm0.51}$ | $0.24_{\pm0.03}$ | $-0.88_{\pm0.27}$ |
| DenseNet40 | $20.8_{\pm1.37}$ | $20.9_{\pm1.22}$ | $20.3_{\pm1.46}$ | $20.6_{\pm1.21}$ | $19.2_{\pm1.18}$ | $17.6_{\pm1.07}$ | $10.3_{\pm0.78}$ | $14.4_{\pm0.91}$ | $7.58_{\pm0.48}$ | $0.38_{\pm0.01}$ | $+0.44_{\pm0.12}$ |
| **PACS** | | | | | | | | | | | |
| $\to$ *Photo* | | | | | | | | | | | |
| ResNet50 | $7.58_{\pm0.37}$ | $7.67_{\pm0.44}$ | $7.43_{\pm0.37}$ | $7.61_{\pm0.42}$ | $7.34_{\pm0.29}$ | $6.52_{\pm0.31}$ | $5.63_{\pm0.19}$ | $2.93_{\pm0.19}$ | $2.64_{\pm0.12}$ | $0.42_{\pm0.04}$ | $+0.46_{\pm0.09}$ |
| DenseNet121 | $8.21_{\pm0.49}$ | $8.37_{\pm0.44}$ | $8.16_{\pm0.49}$ | $8.31_{\pm0.48}$ | $7.97_{\pm0.34}$ | $7.22_{\pm0.36}$ | $5.99_{\pm0.26}$ | $3.54_{\pm0.21}$ | $3.27_{\pm0.18}$ | $0.49_{\pm0.06}$ | $+0.26_{\pm0.04}$ |
| Wide-Res50 | $6.63_{\pm0.33}$ | $6.73_{\pm0.36}$ | $6.53_{\pm0.38}$ | $6.67_{\pm0.34}$ | $6.48_{\pm0.36}$ | $5.89_{\pm0.28}$ | $4.93_{\pm0.22}$ | $2.53_{\pm0.12}$ | $2.27_{\pm0.09}$ | $0.38_{\pm0.01}$ | $+0.31_{\pm0.13}$ |
| $\to$ *Art* | | | | | | | | | | | |
| ResNet50 | $13.1_{\pm0.71}$ | $13.4_{\pm0.64}$ | $12.7_{\pm0.69}$ | $13.2_{\pm0.63}$ | $12.3_{\pm0.57}$ | $10.2_{\pm0.43}$ | $8.26_{\pm0.31}$ | $4.92_{\pm0.28}$ | $4.57_{\pm0.22}$ | $0.76_{\pm0.04}$ | $-0.23_{\pm0.11}$ |
| DenseNet121 | $13.7_{\pm0.76}$ | $14.3_{\pm0.74}$ | $13.7_{\pm0.81}$ | $13.8_{\pm0.69}$ | $12.6_{\pm0.68}$ | $11.2_{\pm0.57}$ | $8.96_{\pm0.46}$ | $5.54_{\pm0.39}$ | $5.63_{\pm0.29}$ | $0.84_{\pm0.09}$ | $-0.18_{\pm0.11}$ |
| Wide-Res50 | $12.2_{\pm0.56}$ | $12.4_{\pm0.54}$ | $12.3_{\pm0.64}$ | $12.8_{\pm0.54}$ | $11.2_{\pm0.57}$ | $9.69_{\pm0.47}$ | $7.58_{\pm0.39}$ | $4.26_{\pm0.23}$ | $3.86_{\pm0.19}$ | $0.63_{\pm0.04}$ | $+0.41_{\pm0.14}$ |
| $\to$ *Cartoon* | | | | | | | | | | | |
| ResNet50 | $16.1_{\pm0.84}$ | $16.7_{\pm0.81}$ | $15.8_{\pm0.94}$ | $16.4_{\pm0.79}$ | $15.4_{\pm0.72}$ | $13.3_{\pm0.62}$ | $10.1_{\pm0.54}$ | $6.54_{\pm0.47}$ | $6.18_{\pm0.34}$ | $1.06_{\pm0.11}$ | $+0.62_{\pm0.24}$ |
| DenseNet121 | $16.9_{\pm0.91}$ | $17.2_{\pm0.99}$ | $16.4_{\pm1.09}$ | $16.9_{\pm0.84}$ | $15.8_{\pm0.82}$ | $14.1_{\pm0.78}$ | $10.6_{\pm0.64}$ | $7.27_{\pm0.54}$ | $6.89_{\pm0.47}$ | $1.16_{\pm0.11}$ | $+0.37_{\pm0.16}$ |
| Wide-Res50 | $15.7_{\pm0.71}$ | $15.6_{\pm0.78}$ | $15.2_{\pm0.86}$ | $15.6_{\pm0.71}$ | $14.3_{\pm0.68}$ | $12.6_{\pm0.58}$ | $9.59_{\pm0.53}$ | $5.81_{\pm0.39}$ | $5.53_{\pm0.33}$ | $0.96_{\pm0.07}$ | $+0.58_{\pm0.13}$ |
| $\to$ *Sketch* | | | | | | | | | | | |
| ResNet50 | $19.2_{\pm1.06}$ | $19.4_{\pm1.12}$ | $18.9_{\pm1.28}$ | $19.1_{\pm1.09}$ | $18.4_{\pm1.09}$ | $15.6_{\pm0.93}$ | $13.4_{\pm0.84}$ | $8.57_{\pm0.76}$ | $8.94_{\pm0.67}$ | $1.31_{\pm0.09}$ | $-0.87_{\pm0.24}$ |
| DenseNet121 | $19.9_{\pm1.12}$ | $20.3_{\pm1.24}$ | $19.6_{\pm1.34}$ | $20.1_{\pm1.16}$ | $18.9_{\pm1.13}$ | $17.1_{\pm1.07}$ | $13.9_{\pm0.98}$ | $9.28_{\pm0.86}$ | $8.62_{\pm0.76}$ | $1.49_{\pm0.16}$ | $+0.28_{\pm0.16}$ |
| Wide-Res50 | $18.8_{\pm0.94}$ | $18.7_{\pm1.07}$ | $18.1_{\pm1.16}$ | $18.6_{\pm0.99}$ | $17.6_{\pm0.93}$ | $14.9_{\pm0.86}$ | $12.6_{\pm0.76}$ | $7.84_{\pm0.63}$ | $7.59_{\pm0.57}$ | $1.24_{\pm0.12}$ | $+0.47_{\pm0.26}$ |
| **I-S** | | | | | | | | | | | |
| $\to$ *Sketch* | | | | | | | | | | | |
| ResNet152 | $22.4_{\pm1.28}$ | $22.6_{\pm1.38}$ | $21.6_{\pm1.48}$ | $22.6_{\pm1.28}$ | $21.1_{\pm1.26}$ | $18.2_{\pm1.11}$ | $14.4_{\pm0.93}$ | $10.1_{\pm0.88}$ | $9.52_{\pm0.76}$ | $1.56_{\pm0.11}$ | $+0.87_{\pm0.32}$ |
| DenseNet161 | $23.2_{\pm1.33}$ | $23.1_{\pm1.43}$ | $22.4_{\pm1.52}$ | $22.8_{\pm1.39}$ | $21.8_{\pm1.33}$ | $18.9_{\pm1.23}$ | $14.8_{\pm1.09}$ | $11.1_{\pm0.96}$ | $10.1_{\pm0.82}$ | $1.66_{\pm0.11}$ | $-0.69_{\pm0.28}$ |
| ViT-L | $12.3_{\pm0.73}$ | $12.4_{\pm0.86}$ | $11.8_{\pm0.88}$ | $12.1_{\pm0.79}$ | $11.3_{\pm0.78}$ | $9.84_{\pm0.68}$ | $7.56_{\pm0.54}$ | $5.28_{\pm0.49}$ | $4.84_{\pm0.36}$ | $0.86_{\pm0.06}$ | $+1.26_{\pm0.24}$ |

*Table 7.* Comparison between Mini-Batch Non-Trainable ECL and Mini-Batch Trainable ECL on Digit and PACS benchmark tasks. Results report ECE (%), CwECE (%), $ECE^{KDE}$ (%) and accuracy change $\Delta$ACC (%) with mean $\pm$ std over five runs. ECE represents the results under top-label calibration, CwECE represents the results under class-wise calibration, and $ECE^{KDE}$ represents the results under canonical calibration.

| Dataset | Architecture | Method | Top-Label | | Class-wise | | Canonical | |
|---|---|---|---|---|---|---|---|---|
| | | | ECE (%) $\downarrow$ | $\Delta$ACC (%) | CwECE (%) $\downarrow$ | $\Delta$ACC (%) | $ECE^{KDE}$ (%) $\downarrow$ | $\Delta$ACC (%) |
| Digit ($\to$ MNIST) | LeNet-5 | Non-Trainable | $8.85_{\pm0.72}$ | $-0.45_{\pm0.25}$ | $1.75_{\pm0.15}$ | $-0.35_{\pm0.15}$ | $1.68_{\pm0.12}$ | $-0.21_{\pm0.10}$ |
| | | Trainable | $\mathbf{8.52}_{\pm0.78}$ | $-0.92_{\pm0.35}$ | $\mathbf{1.66}_{\pm0.12}$ | $-0.44_{\pm0.09}$ | $\mathbf{1.58}_{\pm0.09}$ | $-0.32_{\pm0.08}$ |
| | ResNet20 | Non-Trainable | $8.05_{\pm0.51}$ | $+0.85_{\pm0.32}$ | $\mathbf{1.38}_{\pm0.13}$ | $+0.45_{\pm0.15}$ | $1.32_{\pm0.08}$ | $+0.38_{\pm0.12}$ |
| | | Trainable | $\mathbf{7.88}_{\pm0.45}$ | $+1.25_{\pm0.42}$ | $1.41_{\pm0.11}$ | $+0.62_{\pm0.11}$ | $\mathbf{1.29}_{\pm0.04}$ | $+0.54_{\pm0.12}$ |
| | DenseNet40 | Non-Trainable | $\mathbf{9.05}_{\pm0.65}$ | $+0.42_{\pm0.18}$ | $1.68_{\pm0.15}$ | $+0.12_{\pm0.08}$ | $1.52_{\pm0.11}$ | $+0.09_{\pm0.06}$ |
| | | Trainable | $9.15_{\pm0.61}$ | $+0.68_{\pm0.20}$ | $\mathbf{1.57}_{\pm0.12}$ | $+0.23_{\pm0.11}$ | $\mathbf{1.42}_{\pm0.13}$ | $+0.19_{\pm0.02}$ |
| PACS ($\to$ Photo) | ResNet50 | Non-Trainable | $7.15_{\pm0.38}$ | $+0.32_{\pm0.15}$ | $3.08_{\pm0.18}$ | $+0.28_{\pm0.11}$ | $\mathbf{2.58}_{\pm0.15}$ | $+0.25_{\pm0.10}$ |
| | | Trainable | $\mathbf{6.87}_{\pm0.34}$ | $+0.72_{\pm0.17}$ | $\mathbf{2.92}_{\pm0.12}$ | $+0.48_{\pm0.09}$ | $2.64_{\pm0.12}$ | $+0.46_{\pm0.09}$ |
| | DenseNet121 | Non-Trainable | $6.35_{\pm0.45}$ | $-0.15_{\pm0.21}$ | $3.72_{\pm0.22}$ | $+0.12_{\pm0.09}$ | $3.41_{\pm0.19}$ | $+0.11_{\pm0.08}$ |
| | | Trainable | $\mathbf{5.96}_{\pm0.27}$ | $-0.83_{\pm0.23}$ | $\mathbf{3.56}_{\pm0.19}$ | $+0.29_{\pm0.11}$ | $\mathbf{3.27}_{\pm0.18}$ | $+0.26_{\pm0.04}$ |
| | Wide-Res50 | Non-Trainable | $2.75_{\pm0.15}$ | $+0.41_{\pm0.12}$ | $2.71_{\pm0.12}$ | $+0.15_{\pm0.08}$ | $2.40_{\pm0.11}$ | $+0.14_{\pm0.07}$ |
| | | Trainable | $\mathbf{2.68}_{\pm0.33}$ | $+0.69_{\pm0.22}$ | $\mathbf{2.58}_{\pm0.09}$ | $+0.34_{\pm0.12}$ | $\mathbf{2.27}_{\pm0.09}$ | $+0.31_{\pm0.13}$ |

*Table 8.* Ablation study on the hyperparameter $\gamma$ on Digit and PACS datasets. $\gamma$ controls the non-linear scaling of the loss weight.

| $\gamma$ | Top-Label | | Class-Wise | | Canonical | |
|---|---|---|---|---|---|---|
| | ECE $\downarrow$ | $\Delta$ACC | CwECE $\downarrow$ | $\Delta$ACC | $ECE^{KDE}$ $\downarrow$ | $\Delta$ACC |
| **Digit ($\to$ MNIST) using ResNet20** | | | | | | |
| 0.5 | $8.76_{\pm0.62}$ | $+1.68_{\pm0.33}$ | $1.94_{\pm0.16}$ | $+1.15_{\pm0.22}$ | $1.83_{\pm0.12}$ | $+0.95_{\pm0.18}$ |
| 0.8 | $8.12_{\pm0.54}$ | $+1.45_{\pm0.29}$ | $1.48_{\pm0.14}$ | $+0.88_{\pm0.16}$ | $1.35_{\pm0.09}$ | $+0.72_{\pm0.14}$ |
| 1.0 | $7.88_{\pm0.45}$ | $+1.25_{\pm0.42}$ | $1.41_{\pm0.11}$ | $+0.62_{\pm0.11}$ | $1.29_{\pm0.04}$ | $+0.54_{\pm0.12}$ |
| 1.2 | $7.85_{\pm0.49}$ | $+0.92_{\pm0.25}$ | $1.55_{\pm0.13}$ | $+0.35_{\pm0.09}$ | $1.38_{\pm0.07}$ | $+0.28_{\pm0.08}$ |
| 1.5 | $8.42_{\pm0.56}$ | $+0.45_{\pm0.18}$ | $1.78_{\pm0.15}$ | $+0.12_{\pm0.05}$ | $1.56_{\pm0.10}$ | $+0.08_{\pm0.04}$ |
| **PACS ($\to$ Photo) using ResNet50** | | | | | | |
| 0.5 | $7.45_{\pm0.41}$ | $+0.88_{\pm0.19}$ | $3.25_{\pm0.22}$ | $+0.65_{\pm0.14}$ | $2.98_{\pm0.18}$ | $+0.62_{\pm0.11}$ |
| 0.8 | $7.02_{\pm0.38}$ | $+0.81_{\pm0.17}$ | $3.05_{\pm0.15}$ | $+0.55_{\pm0.12}$ | $2.58_{\pm0.14}$ | $+0.54_{\pm0.10}$ |
| 1.0 | $6.87_{\pm0.34}$ | $+0.72_{\pm0.17}$ | $2.92_{\pm0.12}$ | $+0.48_{\pm0.09}$ | $2.64_{\pm0.12}$ | $+0.46_{\pm0.09}$ |
| 1.2 | $6.95_{\pm0.32}$ | $+0.61_{\pm0.15}$ | $2.98_{\pm0.14}$ | $+0.41_{\pm0.08}$ | $2.68_{\pm0.10}$ | $+0.39_{\pm0.08}$ |
| 1.5 | $7.18_{\pm0.36}$ | $+0.42_{\pm0.12}$ | $3.12_{\pm0.16}$ | $+0.25_{\pm0.06}$ | $2.89_{\pm0.15}$ | $+0.24_{\pm0.07}$ |

