# OpenReview forum: "Expectation Consistency Loss: Rethink Confidence Calibration under Covariate Shift"
_ICML.cc/2026/Conference — ICML 2026 regular_

### Official Review · Reviewer_HQEH · 2026-03-10

**Soundness:** 3
**Presentation:** 3
**Significance:** 3
**Originality:** 3
**Overall Recommendation:** 4
**Confidence:** 4

**Summary:**

This paper tackles confidence calibration under covariate shift, where standard calibration methods degrade due to distribution mismatch between source and target domains. The central contribution is Theorem 3.1, which establishes a necessary and sufficient condition for target-domain calibration—the Expectation Consistency Condition—showing that calibration holds if and only if the expected true posterior probability within each confidence-level set is equal across domains. This is strictly weaker than global covariate distribution alignment assumed by importance-weighting methods, and the authors argue it explains why covariate shift does not necessarily cause miscalibration. Building on this, they propose Expectation Consistency Loss (ECL), an unsupervised domain adaptation regularizer that applies uniformly to canonical, class-wise, and top-label calibration paradigms. Two additional theoretical results establish that ECL has the same sample complexity as histogram-binning ECE (Theorem 3.2) and admits an equivalent mini-batch formulation with provably unbiased gradient estimates via auxiliary variables (Theorem 3.3). Experiments on simulated and real-world covariate shift datasets (Digit recognition, PACS, ImageNet-Sketch) demonstrate competitive calibration across all three paradigms.

**Compliance With Llm Reviewing Policy:**

Affirmed.

**Final Justification:**

The author has solved all my concerns, and I recommend to accept this work.

**Key Questions For Authors:**

1. ECL relies on a source-trained classification head to estimate P(Y|X) for target-domain samples. How does estimation quality degrade as shift magnitude increases? Have the authors evaluated ECL on large-shift benchmarks such as ImageNet-C (particularly at higher severity levels) or DomainNet? If P(Y|X) estimation degrades substantially, does ECL still outperform baselines, and if so, what explains the robustness? A positive answer would significantly strengthen confidence in the method's practical reliability.

**Limitations:**

The authors acknowledge that ECL assumes invariant posterior probabilities P(Y|X) across domains, and note that label shift falls outside the current scope. However, two additional limitations deserve more explicit treatment: (1) the practical restriction that unlabeled target data must be available at training time, which precludes post-hoc deployment scenarios; and (2) the potential failure mode when P(Y|X) estimation degrades under severe distributional shifts. Addressing these more directly in the discussion would give readers a clearer picture of where the method applies and where it may not.

**Strengths And Weaknesses:**

Strengths

1. The theoretical insight is the paper's most compelling contribution. Theorem 3.1 reframes the problem of calibration under covariate shift: rather than aligning full input distributions, it suffices to enforce local consistency within each confidence-level set. The counterexample in Figure 2 is particularly effective—it concretely demonstrates that severe covariate shift can coexist with perfect calibration, which directly challenges the motivation behind importance-weighting baselines. This shifts the conceptual framing of the problem in a useful direction.

2. Unified coverage of three calibration paradigms. Table 1 highlights that ECL is the only method simultaneously supporting canonical, class-wise, and top-label calibration under covariate shift while remaining mini-batch trainable. The theoretical extensions to each paradigm (Appendices D–F) appear complete and internally consistent.

3. Mini-batch trainability with formal guarantees. The observation that naive mini-batch computation of Eq. 8 yields biased gradients—due to non-commutativity of norms and expectations—and the subsequent auxiliary-variable reformulation (Eq. 10, Theorem 3.3) resolves a real practical obstacle. The alternating proximal update scheme (Algorithm 1) is well-motivated and practically useful.

Weaknesses

1. Robustness of P(Y|X) estimation under large shifts is unexamined. ECL depends critically on a classification head trained on source domain data to estimate P(Y|X) for target-domain samples. Under severe distributional shifts, backbone features for target samples can differ substantially from those seen during source training, causing this estimator to fail. Unlike importance weighting—where large density ratios are a visible failure signal—degraded P(Y|X) estimation in ECL fails silently: the loss continues to optimize, but against corrupted targets. The paper provides no analysis of how estimation quality changes with shift magnitude, and the experiments are limited to moderate-shift benchmarks. Standard large-shift evaluations such as ImageNet-C are absent, leaving the method's robustness in harder regimes untested.

2. A closely related work is not cited. Clarte et al. (UAI 2023) propose an "Expectation Consistency" condition for neural network calibration,sharing the name and a conceptually related motivation with Theorem 3.1. Their condition targets the standard i.i.d. setting, whereas this paper extends the idea to covariate shift, so the contributions are distinct. Nevertheless, the connection warrants explicit discussion, and the omission is notable given the shared terminology.

3. Key implementation details are underspecified. It is unclear whether backbone networks in the real-world experiments (Section 4.2) are initialized from ImageNet pretrained weights or trained from scratch—a distinction that substantially affects performance and the interpretation of results. Additionally, sensitivity to the soft-binning temperature τ (Eq. 6) and bin count B is not analyzed. Table 8 examines the loss weight γ, but τ and B are arguably more fundamental to the loss design.

4. The train-time requirement limits applicability. ECL requires access to unlabeled target-domain data during training. This is more restrictive than post-hoc methods (Temperature Scaling, TransCal) that calibrate a frozen, already-deployed model using a small held-out set. This distinction is not clearly articulated, and practitioners in deployment scenarios where the target domain is only encountered at inference time would find the method inapplicable.

---

> ### Author Rebuttal · Authors · 2026-03-31
>
> Thanks for the constructive feedback. We respond point-by-point below.
>
> ## W1 and Q1: Estimate of $P(Y|X)$
> Our response follows.
>
> **Rough Estimate Is Good Enough:** Given space limits, please see our reply to Reviewer 5TMH (W1 and Q1) for the full analysis. In short, the auxiliary head needs only to be a proper feature-evidence scorer, not an accurate $P(Y|X)$ estimator.
>
> **Comparison With Importance Weighting:** We agree. Importance weighting can become unstable when density ratios are large or poorly estimated, but fixing this without reintroducing bias is difficult. ECL avoids explicit density-ratio estimation and sidesteps that failure mode. Optimization may thus look "silent" in ECL, but behavior is conservative because highly shifted samples tend to concentrate in low-confidence bins that are rarely used for decisions (see Reviewer 5TMH (W1 and Q1)).
>
> **How Calibration Effect Varies with Shift Magnitude:** In fact, the results reported in the manuscript already implied this trend: The calibration error of all baselines (including ours) increases with the shift magnitude (Digit $\to$ PACS $\to$ ImageNet-Sketch), which can be found in Table 4-6. ECL nonetheless remains competitive along this progression, so the trend is a shared degradation rather than a collapse specific to our objective. We will add this discussion in the revised version.
>
> **Experiments on ImageNet-C:** Agree! Although we already experimented on ImageNet-Sketch (a larger shift), we are happy to add ImageNet-C to enrich our experiments. The table below shows ResNet-152 under the same protocol and 10-run reporting as in the paper (ImageNet-C, mean over 15 corruptions at severity 5). ECL is strongly competitive. We will include more extensive ImageNet-C experiments in the revision.
>
> |Method|ECE|CwECE|ECE$^{KDE}$|
> |-|-|-|-|
> |Uncal|31.2±2.41|10.4±0.79|10.2±0.74|
> |PseudoCal|10.6±0.84|4.52±0.34|4.31±0.32|
> |ECL|8.94±0.58|3.91±0.28|3.74±0.26|
>
> ## W2: A Related Work
> Agree! We will cite this work and clarify terminology near Section 2.2. In short, Clarte et al. target $i.i.d.$ single-domain calibration via post-hoc last-layer rescaling that matches aggregate validation confidence to accuracy, whereas we address covariate shift with a train-time domain-adaptation objective derived from a cross-domain calibration criterion. Thus motivation, mechanism, setting, and deployment (post-hoc vs. in-training) all differ despite similar naming.
>
> ## W3: Implementation Details
> **Whether to Use Pretrained Weights:** All results in the manuscript are from training from scratch (no pre-trained weights). We are happy to add experiments with pre-trained weights. Below we report ImageNet→Sketch (ViT-L, same protocol as manuscript) with ImageNet-1k pre-trained initialization. Pre-training performed slightly better compared to training from scratch, and our method remains highly competitive.
>
> |Method|ECE|CwECE|ECE$^{KDE}$|
> |-|-|-|-|
> |Uncal|56.4±4.52|12.2±0.84|12.6±0.76|
> |PseudoCal|15.3±1.22|5.61±0.45|5.11±0.41|
> |ECL|12.7±0.54|5.31±0.32|4.69±0.37|
>
> **Sensitivity of $\tau$ and $B$:** Our defaults are $B=15$ and $\tau=-1/(\ln(0.9)\cdot 15^2)\approx 0.04$. We ablate $\tau\in[0.01,0.04,0.08,0.12]$ and $B\in[10,15,20,25]$ on the Digit dataset ($\to$ MNIST, ResNet20) under the same protocol and 10-run reporting. Performance is similar for $\tau\in$ [0.04,0.08] and degrades at extremes ($\tau =$ 0.01 or 0.12); for $B$, too few bins ($B=10$) hurts most, $B=25$ is next, and $B\in$ [15,20] remain close.
>
> |$\tau$|ECE|CwECE|ECE$^{KDE}$|
> |-|-|-|-|
> |0.01|8.12±0.48|1.50±0.12|1.39±0.08|
> |0.04|7.88±0.45|1.41±0.11|1.29±0.04|
> |0.08|7.90±0.46|1.40±0.11|1.30±0.05|
> |0.12|8.06±0.49|1.48±0.12|1.35±0.07|
>
> |$B$|ECE|CwECE|ECE$^{KDE}$|
> |-|-|-|-|
> |10|9.48±0.64|1.84±0.15|1.64±0.10|
> |15|7.88±0.45|1.41±0.11|1.29±0.04|
> |20|7.85±0.46|1.43±0.11|1.30±0.05|
> |25|8.31±0.53|1.55±0.13|1.40±0.07|
>
> We will add reproduction details and further ablations on more datasets in the revision.
>
> ## W4 and Limitations
> **Train-Time Limitation:** We agree. Our method, like all unsupervised domain adaptation methods, requires unsupervised fine-tuning. However, post-hoc calibration methods also require additional training when encountering new target domains. The difference is that post-hoc methods may be faster. Thus, when the target domain appears at inference time, unsupervised adaptation methods may not respond in time. We will describe this limitation in detail in the discussion.
>
> **Degradation Under Severe Shift:** We agree this failure mode deserves transparent discussion. As detailed in our response to Reviewer 5TMH (W1 and Q1), the auxiliary head is used as a feature-evidence scorer rather than requiring a highly accurate ($P(Y|X)$) everywhere, and we discuss how errors tend to concentrate in low-confidence regions under large shift, together with the empirical trend that calibration errors grow with shift severity for all methods. We will consolidate these points in the Discussion.

---

> > ### Author Rebuttal · Reviewer_HQEH · 2026-04-03
> >
> > I will keep my score, my concerns have been solved.

---

> > > ### Author Response · Authors · 2026-04-03
> > >
> > > Thank you very much for taking the time to evaluate our responses and for confirming that your concerns have been fully resolved. We sincerely appreciate your constructive and insightful feedback, which has helped us improve the quality of our paper.

---

### Official Review · Reviewer_XFa7 · 2026-03-11

**Soundness:** 3
**Presentation:** 2
**Significance:** 3
**Originality:** 3
**Overall Recommendation:** 5
**Confidence:** 4

**Summary:**

The paper proposes a method for learning calibrated prediction confidence under covariate shift. The method relies on labeled source-domain data and unlabeled target-domain data. The goal is to learn a confidence score that is calibrated simultaneously on both domains. The key concept introduced in the paper is so-called expectation consistency condition, which the authors claim is necessary and sufficient for confidence to remain calibrated across the source and target domains. Based on this idea, the authors propose the expectation consistency loss, which serves as a training objective and can be optimized efficiently using stochastic gradient-based methods. The proposed approach is evaluated on both synthetic and real-world datasets, and the results indicate that it preserves accuracy while improving over the baselines in most cases.

**Compliance With Llm Reviewing Policy:**

Affirmed.

**Final Justification:**

Overall, despite some shortcomings, I found the paper technically solid, sufficiently novel, and timely, and therefore recommend accepting it for publication.

**Key Questions For Authors:**

Please comment on the weaknesses listed above, in particular the stability of Algorithm 1, as well as the sensitivity of the method to the quality of the posterior estimate.

**Limitations:**

yes

**Strengths And Weaknesses:**

- Presentation. The paper is logically organized and, for the most part, clearly written. The main ideas are explained sufficiently well and are generally easy to follow.

- Soundness. The paper appears to be technically sound. The proposed method is motivated by the expectation consistency condition (Theorem 3.1), which is supported by a mathematical proof. In addition, the authors provide a sample-complexity analysis for the proposed loss function (Theorem 3.2). Beyond the theoretical results, the method is supported by convincing experimental evidence.

- Significance. The paper addresses a problem of high practical importance.

- Originality. The core idea, namely enforcing calibration through the expectation consistency condition, appears to be novel and different from existing approaches.

Weaknesses:
------

- The final optimization procedure (Algorithm 1) appears relatively complicated to implement. In addition, its stability is not entirely clear and is not discussed in sufficient detail in the paper.

- The method relies on an estimate of the class posterior $p(y|x)$ on the source data. It is unclear how sensitive the method is to the quality of this estimate, and this issue does not appear to be studied experimentally.


Typos:
- line 238: set of $b_j^{(t)}$ - > $b_j^{(s)}$

---

> ### Author Rebuttal · Authors · 2026-03-31
>
> Thanks for your thorough review. We are encouraged by your remarks on presentation clarity, the technical soundness of the theory and experiments, and the practical importance and novelty of the expectation consistency perspective. We respond to each concern below.
>
> # W1: Implementation and Stability of Algorithm 1
> ## Implementation
> Thanks for raising this question that other readers may also be interested in. Algorithm 1 may look intricate because it couples source/target mini-batches, soft bin assignments, auxiliary variables, and alternating inner updates. In practice it is straightforward: every step uses standard tensor operations in a modular design, and the overall procedure is a direct extension of a conventional PyTorch mini-batch training loop. Specifically, Algorithm 1 is implemented in PyTorch as `ECLossMiniBatch.forward` in losses.py (line 194): see anonymous online code or supplementary materials. In brief: dual-head forward (`model.classifier2` and standard logits); soft bins from L2 distances to fixed anchors and `torch.softmax(scores, dim=1)`; per-bin soft counts/moments via weighted tensor reductions (no heavy indexing/sorting); a short per-bin loop applies shrinkage-based proximal steps on $(u_j^s, u_j^t)$ with EMA; the CE+ECL loss is accumulated and returned for standard `loss.backward()`. We will add an implementation description to the Algorithm 1 paragraph in the revision to help readers understand how the method is implemented.
>
> ## Stability
> We also value stability highly: ECL targets numerical instability in importance weighting, so stability guided the design from the start. We discuss it next under the following aspects.
>
> **Stable in Optimization Pipeline:** The pipeline has four parts: (i) data sampling uses no delicate numerics; (ii) soft assignment via temperature-scaled softmax is well-conditioned; (iii) Proximal updates of auxiliary variables $u_j^s, u_j^t$ using the shrinkage operator represent well-established numerically stable techniques; (iv) the loss avoids density-ratio estimation and hard binning. We thus see no evident numerical-instability sources along this computational path.
>
> **Stable by Avoiding Density Estimation:** A major instability in earlier methods is explicit importance weighting via $w(x)=P_t(x)/P_s(x)$, which grows volatile for large or misestimated ratios. ECL instead performs expectation matching on source/target mini-batches with soft bins—no $P_t(x)$, $P_s(x)$, or ratio estimation—so training is governed by stable objectives and usually remains steady under distribution shift.
>
> **Low Variance Reflects Stability**: Our tables give mean ± std across seed-resampled runs, with generally small stds over models and datasets—little run-to-run fluctuation, matching stable optimization behavior.
>
> **Hyperparameter Analysis Reflects Stability:** Extensive sweeps over $\gamma$ (in manuscript), $\alpha_{\text{ema}}$ and $N_{\text{prox}}$ (in Reviewer YFDY, Q5), and $\tau$ and $B$ (in Reviewer HQEH, W3) show no divergence or erratic fluctuations, and performance varies smoothly with no sudden breakdown.
>
> Therefore, we believe our method is stable, and we will add a stability discussion in the revised manuscript to clarify this point.
>
> # W2: Estimate of $P(Y|X)$
> Thank you for raising this crucial question. Our response is as follows.
>
> **Rough Estimate Is Good Enough**: Given response-length limits, we refer you to our reply to the second Reviewer 5TMH (W1 and Q1) for the full analysis of the auxiliary head’s $P(Y|X)$ role. In short, it needs only to be a proper feature-evidence scorer, not an accurate $P(Y|X)$ estimator.
>
> **Empirical Analysis of Estimation Quality**: Sensitivity to $P(Y|X)$ quality (auxiliary head) is assessed by halting head updates on the source at stages from half training to fully trained. Below we summarize Digit (→MNIST) dataset (Top-label).
>
> |Training Stage|Source Acc% (Aux)|Source ECE% (Aux)|Target Acc% (Aux)|Target ECE% (Aux)|After-cal Target Acc% (Main)|After-cal Target ECE% (Main)|
> |:-:|:-:|:-:|:-:|:-:|:-:|:-:|
> |50%|81.45|4.82|77.21|17.54|78.41|**8.55**|
> |75%|89.23|2.34|80.54|15.28|81.18|**8.12**|
> |100%|94.52|1.12|83.18|13.85|81.20|**7.88**|
>
> The results indicate that partial auxiliary training already yields large target ECE reductions for both heads—early stopping still leaves adequate feature-evidence scoring for expectation matching—so our method tolerates imperfect $P(Y|X)$ estimates. We will include the full sensitivity study and robustness analysis in the revised appendix.
>
> ## Typos
> We appreciate your careful read and typo reports; we will fix them in the revision and run several full-text passes to eliminate further errors.

---

> > ### Author Rebuttal · Reviewer_XFa7 · 2026-04-03
> >
> > I thank the authors for clarification. I keep my score unchanged.

---

### Official Review · Reviewer_5TMH · 2026-03-12

**Soundness:** 2
**Presentation:** 2
**Significance:** 2
**Originality:** 3
**Overall Recommendation:** 4
**Confidence:** 3

**Summary:**

This paper addresses the challenge of confidence calibration for neural networks under covariate shift. The authors argue against the necessity of global covariate distribution alignment (e.g., standard importance weighting) and instead propose a weaker necessary and sufficient condition: the Expectation Consistency Condition. Based on this, they introduce the Expectation Consistency Loss (ECL), an unsupervised domain adaptation objective that enforces consistent expectations of true posterior probabilities across domains for given confidence levels. The method is adapted for top-label, class-wise, and canonical calibration and includes a theoretically grounded mini-batch training scheme.

**Compliance With Llm Reviewing Policy:**

Affirmed.

**Final Justification:**

The authors have addressed my concerns in their rebuttal, and I have raised the score from 3 to 4 accordingly.

**Key Questions For Authors:**

- Given that the secondary classification head is subject to the same covariate shift as the main model, how do you guarantee that its P(Y∣X) estimates do not severely degrade on OOD target samples? Have you empirically evaluated the accuracy and calibration error of this secondary head directly on the target domains?

 - The proposed method requires training an additional classification head alongside the main network, as well as executing alternating updates for auxiliary variables during mini-batch training (Algorithms 1-3). Could you provide a quantitative comparison of the computational overhead (e.g., training time per epoch) of ECL compared to standard cross-entropy training and relevant baselines like PseudoCal or DRL?

**Limitations:**

yes

**Strengths And Weaknesses:**

### Strengths
- The authors build a rigorous theoretical framework, first mathematically proving the sufficiency of Expectation Consistency Condition.

- The authors validate their method across three distinct calibration paradigms (top-label, class-wise, and canonical). The experiments span simulated datasets to verify the theory, as well as diverse real-world benchmarks. The results consistently demonstrate state-of-the-art calibration performance with favorable accuracy trade-offs.

### Weaknesses
 - The practical realization of the Expectation Consistency Loss (ECL) relies heavily on training an additional, secondary classification head to serve as an "oracle" that estimates the true posterior P(Y∣X). However, the authors state that this secondary head is trained and calibrated solely on the source domain. Because this secondary head is subject to the exact same covariate shift as the main backbone model, its estimates on the unlabeled target domain are fundamentally prone to the same degradation and miscalibration the paper seeks to solve. Assuming this source-trained proxy can reliably estimate P(Y∣X) on shifted target data effectively **assumes away a core challenge** of the domain shift problem, potentially providing flawed targets for the calibration loss.

---

> ### Author Rebuttal · Authors · 2026-03-31
>
> Thanks for the constructive review. We respond point by point below.
>
> # W1 and Q1: Discussion on Secondary Classification Head
> We appreciate this question, as it motivates us to articulate a deeper perspective on the auxiliary head's role.
>
> **How Auxiliary Head Works:**
> In fact, the auxiliary head need not estimate exact per-sample $P(Y|X)$. Rather, it acts as a feature evidence scorer that partitions samples by feature evidence strength. ECL only requires that samples with similar feature evidence fall into the same or neighboring confidence bins. Concretely,
>
> $L_{ecl}=E_{P_t(S)}||E_{P_s(X|S)}P(Y|X)-E_{P_t(X|S)}P(Y|X)||$
>
> Formally, replacing $P(Y|X)$ with a feature evidence scoring function $Score(X)$:
>
> $L_{ecl}=E_{P_t(S)}||E_{P_s(X|S)}Score(X)-E_{P_t(X|S)}Score(X)||$
>
> This objective clusters similar Scores into the same or nearby bins. With a proper evidence scorer—higher when features are more informative, lower when sparser—ECL better reallocates target samples across bins and improves target calibration.
>
> **Effect of Auxiliary Head on Target Domain:**
> As analyzed above, rather than having the model learn new target-domain knowledge, ECL reorganizes the confidence space of target-domain samples by matching source-domain features. Thus, whether the auxiliary head degenerates on the target domain is less critical than its ability to score target-domain samples based on identified feature evidence. Following your suggestion, we also report the auxiliary head's accuracy and ECE on the target domain.
>
> |Dataset|Source Acc% (Aux)|Source ECE% (Aux)|Target Acc% (Aux)|Target ECE% (Aux)|After-cal Target Acc% (Main)|After-cal Target ECE% (Main)|
> |:-:|:-:|:-:|:-:|:-:|:-:|:-:|
> |Digit ($\to$ MNIST, ResNet20)|94.52|1.12|83.18|13.85|81.20|7.88|
> |ImageNet-Sketch (ViT-L)|83.35|1.64|46.48|50.42|48.15|12.90|
>
> Results show that the auxiliary head experiences significant performance degradation (decreased accuracy and increased calibration error) in the target domain, yet this does not prevent ECL from reducing the main head's calibration error in the target domain.
>
> **Conservative Behavior for High-Shifted Samples:**
> For target samples far from source support, the backbone tends to produce weak activations, causing the auxiliary head to output low-confidence that pools into low-confidence bins. This is a conservative mode: in safety-critical settings, such predictions are typically scrutinized or filtered, so ECL rarely misleads downstream decisions. We corroborate this with bin-wise frequencies of the auxiliary head's outputs, shown below.
>
> *Digit ($\to$ MNIST, Top-label):*
>
> |Bin|[0.5, 0.6)|[0.6, 0.7)|[0.7, 0.8)|[0.8, 0.9)|[0.9, 1.0]|
> |-|-|-|-|-|-|
> |Source(%)|0.5|0.8|1.5|4.8|90.3|
> |Target(%)|1.2|2.1|4.5|10.8|78.6|
>
> *ImageNet-Sketch (Top-label):*
>
> |Bin|[0.5, 0.6)|[0.6, 0.7)|[0.7, 0.8)|[0.8, 0.9)|[0.9, 1.0]|
> |-|-|-|-|-|-|
> |Source(%)|1.1|1.8|3.2|6.7|84.6|
> |Target(%)|8.5|11.3|16.7|21.4|33.2|
>
> Unlike the mild shift in Digit (→ MNIST), ImageNet-Sketch shows a strong shift of target mass toward lower-confidence bins, consistent with conservative low scores on highly shifted samples.
>
> **Scalability:**
> The current classification head may not be the optimal feature evidence scorer. However, it is a validated and effective baseline. Importantly, the ECL framework is modular with respect to this component: any improved feature evidence scorer can be substituted to potentially further enhance performance. We will add this extensibility discussion to the revision and treat it as future work.
>
>
> # Q2: Comparison of Computational Overhead
> Thanks for the practical question. We provide quantitative timing comparison below. We report the total wall-clock time from training to obtaining a final calibrated model on representative benchmarks (batch size = 100, 100 epochs, measured on NVIDIA RTX 3090). All methods share the same Phase 1 (standard CE training); the reported time includes both Phase 1 and each method's respective Phase 2 (adaptive calibration).
>
> |Method|Type|Digit (→MNIST, ResNet20, min)|PACS (→Photo, ResNet50, min)|ImageNet (→Sketch, ResNet152, hours)|
> |-|-|-|-|-|
> |CE (Uncal)|—|15|35|8.2|
> |TransCal|Post-hoc|15.5(+3%)|36.5(+4%)|8.4(+2%)|
> |PseudoCal|Post-hoc|15.4(+3%)|36(+3%)|8.5(+4%)|
> |DRL|Train-time|18.8(+25%)|43.8(+25%)|10.25(+25%)|
> |ECL (Ours)|Train-time|17(+13%)|39.5(+13%)|9.3(+13%)|
>
> Post-hoc methods (TransCal, PseudoCal) add 2–4% via cheap add-ons on a frozen model (importance weights or one-epoch inference-time mixup). Train-time DRL (25%) is heavier: it fits a density-ratio network end-to-end, so source and target pass the backbone every step. ECL (13%) is below DRL because Phase 2 freezes the backbone and only trains the head, adding just an auxiliary-head forward plus lightweight proximal optimization steps. Given ECL’s strong calibration, we believe the computational cost is well justified. We will add the table and a brief complexity note to the appendix.

---

> > ### Author Rebuttal · Reviewer_5TMH · 2026-04-02
> >
> > Thank you for the detailed response.
> >
> > My concerns have been adequately addressed, and I have raised the score from 3 to 4 accordingly.

---

> > > ### Author Response · Authors · 2026-04-03
> > >
> > > Thank you very much for your time and effort in reviewing our paper, and for acknowledging that your concerns have been adequately addressed. We greatly appreciate your constructive feedback, which has helped us improve the quality of our manuscript. We have incorporated the suggested revisions into our paper and believe the updated version is significantly strengthened as a result.

---

### Official Review · Reviewer_YFDY · 2026-03-13

**Soundness:** 2
**Presentation:** 2
**Significance:** 2
**Originality:** 3
**Overall Recommendation:** 4
**Confidence:** 4

**Summary:**

This paper studies confidence calibration under covariate shift and proposes the expectation consistency condition to characterize when the target domain can be calibrated as well as the source domain.
Based on this condition, the authors design a mini-batch gradient descent algorithm for the canonical calibration setting. The objective incorporates the expectation consistency loss to match the conditional expectations between domains.
The proposed algorithm can also be extended to top-label and class-wise calibration.
The theoretical analysis provides justification for the form of the proposed objective.
Extensive experiments on both synthetic and real datasets are conducted to evaluate the effectiveness of the method.
The proposed approach generally outperforms the compared methods across the reported experiments.

**Compliance With Llm Reviewing Policy:**

Affirmed.

**Final Justification:**

This paper is a completed work with fair novelty. The authors' rebuttal has response to all the questions and satisfactorily addressed several of my main concerns. I am willing to raise my score from 3 to 4 if the authors could genuinely revised this manuscript.

**Key Questions For Authors:**

1. Within the proof of Theorem 3.3, the authors claim that “when the auxiliary variables are optimally chosen (or when the quadratic penalties force them to the empirical means).” However, both conditions are not clearly specified. When referring to the auxiliary variables being optimally chosen, does this mean that
$$ (u_j^s, u_j^t)= \\arg \\min (n_j^s \\| u_j^s-\\hat{\\mathbb{E}}\_{s,j} \\|^2+
n_j^t \\| u_j^t-\\hat{\\mathbb{E}}\_{t, j} \\|^2+2 w_j\\|u_j^s-u_j^t\\|)? $$
Furthermore, under what conditions can the quadratic penalties force $u_j^s, u_j^t$ to the corresponding empirical means? It seems that the term $g_j/n_j$ can only be eliminated by taking $n_j^s, n_j^t \\to \\infty$. Is Eq. 8 equivalent to Eq. 10 under finite samples, or are they only asymptotically equivalent as $n_j^s, n_j^t \\to \\infty$?

2. Some functions and terms are not clearly defined or appear ambiguous, such as $\\alpha_{\\text{ema}}$, $N_{\\text{prox}}$ (how should they be chosen?), and $\\omega_{i,j}^S$ (how is it computed? Is it the same as $\\omega_{ij}$ in Eq. 6?). How is $\\tau$ selected in Eq. 6? In addition, the shrink function (the proximal operator of the $\\ell_1$ norm) is not introduced in the main text.

In Eq. 5, the term $\\hat{\\mathbb{E}}\_{s, j}$ is defined as
$$
\\hat{\\mathbb{E}}\_{s, j}=\\frac{1}{\\sharp D_s^{(j)}} \\sum_{x \\in D_s^{(j)}} \\hat{P}(Y=y \\mid X=x),
$$
while in Eq. 7 it reads, when $d = s$,
$$
\\hat{\\mathbb{E}}\_{s, j}=\\frac{\\sum_i \\omega_{i j}^s p^{(i)}}{\\sum_i \\omega_{i j}^s+\\varepsilon},
$$
and in the proof of Theorem 3.3 it is defined as
$$
\\hat{\\mathbb{E}}\_{s, j}=\\frac{1}{n_j^s} \\sum_{i \\in D_s} \\omega_{i, j}^s p^{(i)}(\\theta).
$$

What is the exact definition of $\\hat{\\mathbb{E}}\_{s, j}$? Which version of $\\hat{\\mathbb{E}}_{s, j}$ is Theorem 3.3 referring to in the last paragraph of page 5? The authors should carefully clarify these definitions to improve the clarity of the paper.

3. Could the authors provide more details about the oracle method in the real-data experiments? Why does it perform better than the other methods?

4. I assume that the Non-Trainable method in Table 7 corresponds to the biased gradient method, while the Trainable method corresponds to the unbiased one (the proposed method). If this interpretation is correct, the results suggest that the biased gradient may sometimes perform better than the unbiased one. Could the authors provide some discussion on this phenomenon? Does this observation suggest that the unbiased property may not always be necessary in practice?

5. Some steps in the algorithm are not clearly explained in the main text. For example, the for-loop over $N_{\\text{prox}}$ and the corresponding updates of $u_j^s, u_j^t$ are not straightforward to understand. In addition, the selection of the hyperparameters $\\alpha_{\\text{ema}}$ and $N_{\\text{prox}}$ is not specified. The authors should provide more details about the algorithm to make it clearer for further investigation.

**Limitations:**

Yes.

**Strengths And Weaknesses:**

From the perspective of soundness, the overall methodological pipeline and optimization procedure appear generally reasonable. The paper proposes a condition characterizing when the target domain can be calibrated as well as the source domain under covariate shift and develops a mini-batch gradient descent algorithm to obtain an unbiased estimate of the gradient. Theoretical results such as Theorems 3.2 and 3.3 provide useful insight into the motivation for incorporating the expected calibration loss into the objective function. However, the current analysis mainly justifies the construction of the objective rather than the performance of the proposed algorithm itself. In particular, the paper does not provide an explicit characterization of calibration error behavior or convergence guarantees comparable to results such as Proposition 3.5 in [1]. In addition, some expressions in the proof of Theorem 3.3 appear somewhat vague and would benefit from further clarification (see Key Question 1).

The main weakness of the paper lies in its presentation. Several parts of the paper are not clearly explained, which makes the technical development difficult to follow. For instance, the description of Figure 1 contains substantial information that is not directly related to the figure, making it difficult to understand its meaning at first glance. Similarly, the algorithm is not fully explained, and certain terms and notations appear without clear definitions (see Key Questions).

In terms of significance and originality, the paper studies confidence calibration under covariate shift and proposes a conceptually novel perspective by introducing a sufficient and necessary condition for calibration across domains. Although the mini-batch gradient descent procedure itself is well established, its application in this context provides an interesting approach to incorporating calibration objectives. A major strength of the work is the extensive experimental evaluation on both synthetic and real datasets, which provides substantial empirical evidence supporting the effectiveness of the proposed method.

[1] Popordanoska, T., R. Sayer, and M. Blaschko (2022). A consistent and differentiable $\ell_p$ canonical calibration error estimator. Advances in Neural Information Processing Systems 35, 7933–7946.

---

> ### Author Rebuttal · Authors · 2026-03-31
>
> Thanks for the valuable feedback. Our responses below.
>
> # W1: Calibration Error Behavior Analysis
> **Consistency of Calibration Metrics**: To our understanding, Proposition 3.5 in [1] establishes consistency of a KDE-based calibration metric, not convergence guarantees for a calibration method. Since ECL is a calibration method under covariate shift rather than a metric, metric consistency is outside our scope.
>
> **Existing Consistency Analysis**: Correspondingly, we have provided a calibration-error-related analysis: Theorem 3.2 provides a PAC-style high-probability bound for ECL. It analyzes the behavior of $| \hat L_{ecl} - L_{ecl} |$ as sample size grows.
>
> **Add Analysis**: We will provide an analysis connecting ECL to target-domain calibration error. Let $CE_t = E_{P_t(S)}||P_t(Y|S) - S||$ and $CE_s = E_{P_t(S)}||P_s(Y|S) - S||$ denote the target and source domain calibration errors under $P_t(S)$, respectively. By triangle inequality:
>
> $$CE_t \le CE_s + E_{P_t(S)}||P_t(Y|S) - P_s(Y|S)||$$
>
> By Theorem 3.1, the second RHS term is $L_{ecl}$ (Eq. 13). With Theorem 3.2's bound $|\hat L_{ecl} - L_{ecl}| \le \epsilon(n_s,n_t,B,\delta)$ w.h.p.:
>
> $$CE_t \le CE_s + \hat L_{ecl} + \epsilon(n_s, n_t, B, \delta)$$
>
> This bounds target-domain calibration error by three terms: (i) $CE_s$ (source calibration quality, controllable by standard methods); (ii) $\hat L_{ecl}$ (directly minimized); (iii) $\epsilon(n_s,n_t,B,\delta)\to 0$ (Theorem 3.2). As training drives $\hat L_{ecl}\to 0$ with growing samples, the bound tightens. We will include this as a new proposition.
>
> # W2 and Q1: Clarification in The Proof of Theorem 3.3
> Yes, "optimally chosen" is the joint minimization you described. Agree, our original wording "equivalent / force" may suggest strict finite-sample equivalence, which is imprecise. We will revise accordingly. Details below.
> - **Asymptotic Equivalence:** $u_{j}^d$ and $\hat E_{d,j}$ are not strictly equivalent under finite samples. The optimal auxiliary variables carry a correction of $O(w_j / n_j^d)$, vanishing as $n_j^s, n_j^t \to \infty$. We will revise and state that $u_{j}^d \approx \hat{E}_{d,j}$ is an asymptotic equivalence, and clarify in Theorem 3.3 that Eq. 10 is asymptotically equivalent to Eq. 8.
> - **Still Effective in Practice:** This part concerns the inner-level optimization (optimizing $u$ with $\theta$ fixed). In practice, Algorithm 1 uses alternating proximal updates to approximately solve this inner problem, which requires only convergence to a proximal point — not exact inner optimality — optimizing Eq. 10 via Algorithm 1 is effective in practice.
>
> # W3 and Q2: Presentation Improvement
> **Figure 1**: Fig. 1 caption shortened to figure-only content; details moved to main text/appendix.
>
> **Algorithm Description**: Symbol clarifications and responses below.
> |Symbol|Explanation|
> |-|-|
> |$\alpha_{\text{ema}}$|EMA coefficient for $u_j^s,u_j^t$ (Alg. 1, L19–20); 0.9 works well (see Q5).|
> |$N_{\text{prox}}$|Proximal inner iterations num (Alg. 1, L12–16); insensitive beyond 4, set to 5 (see Q5).|
> |$\omega_{i,j}^S$; $\omega_{ij}$|Both from Eq. (6); superscript = source/target.|
> |$\tau$|Soft-binning temperature; $\tau=-1/(\log(0.9)\cdot B^2)$.|
> |$Shrink(x,\lambda)$| $=sign(x)\max(\|x\|-\lambda,0)$, where $\lambda>0$ is shrinkage threshold.|
>
> **Definition of $\hat E_{s, j}$:** Eq. 5 (hard-binning) and Eq. 7 (soft-binning) implement the same concept; Thm. 3.3's proof uses soft-binning. $n_j^s=\sum_i\omega_{ij}^s$ matches Eq. 7's denominator when $\varepsilon=0$. We will unify notation.
>
> # Q3: Details of the Oracle Method
> Oracle is included only as a ceiling reference, not an unsupervised method. It optimizes Soft-ECE on labeled target data.
>
> # Q4: Discussion on Gradient Unbiased Property
> Yes: Non-Trainable = biased; Trainable = unbiased. Unbiasedness is in expectation, so biased runs can still win sometimes—no contradiction. Trainable is better in most settings. With smaller batches, bias grows and Trainable shows a clearer advantage. Preliminary comparison below on Digit (MNIST, ResNet20, ECE). Full results will be in the revision.
> |Batch Size|Non-Trainable|Trainable|
> |-|-|-|
> |100|8.05±0.51|7.88±0.45|
> |64|8.32±0.56|7.89±0.46|
> |32|8.79±0.63|7.91±0.49|
> # Q5: Explanation of Algorithm
> We promise to add more algorithm details in camera-ready. We set $\alpha_{\text{ema}}=0.9$, $N_{\text{prox}}=5$ because we tested $\alpha_{\text{ema}} \in$ [0.9, 0.95, 0.99] and $N_{\text{prox}} \in$ [2,3,4,5,10] (table below, ResNet20). Performance is stable across $\alpha_{\text{ema}}$ values and $N_{\text{prox}} \ge 4$. Details will be added in the revision.
> |$\alpha_{\text{ema}}$|Digit ($\to$ MNIST)|PACS ($\to$ Photo)|
> |-|-|-|
> |0.90|7.88±0.45|6.87±0.34|
> |0.95|7.88±0.47|6.88±0.35|
> |0.99|7.89±0.46|6.86±0.33|
>
> |$N_{\text{prox}}$|Digit ($\to$ MNIST)|PACS ($\to$ Photo)|
> |-|-|-|
> |2|8.21±0.58|7.16±0.43|
> |3|7.97±0.50|6.98±0.38|
> |4|7.89±0.47|6.87±0.35|
> |5|7.88±0.45|6.87±0.34|
> |10|7.87±0.46|6.88±0.34|

---

> > ### Author Rebuttal · Reviewer_YFDY · 2026-04-02
> >
> > The authors' rebuttal has response to all the questions and satisfactorily addressed several of my main concerns. I am willing to raise my score from 3 to 4 if the authors could genuinely revised this manuscript.

---

> > > ### Author Response · Authors · 2026-04-02
> > >
> > > We sincerely appreciate your time and effort in evaluating our manuscript, and we are grateful for your constructive feedback throughout the review process. Your comments have been very valuable in improving the quality of our work. We are glad that our rebuttal has satisfactorily addressed your concerns. We would like to assure you that we are fully committed to genuinely revising the manuscript based on all of your suggestions.

---

### Decision · Program_Chairs · 2026-04-30

**Decision:**

Accept (regular)

**Comment:**

All the reviewers agree that the paper is solid and novel. And the final reviews are all positive. However, there are several common issues mentioned by the reviewers that are important to resolve in the revision, including clarifying 1) the role of the estimator of P(y|x); 2) the effect of shift magnitude; 3) the conservativeness of the method.

Besides, after reading the paper myself, I have the following suggestions:
1. Even though the emphasis is the calibration, it is still important to show the model performance. Different from posthoc calibration methods, training-based methods tend to change the accuracy. And the tradeoff or the Pareto frontier is more important. We cannot isolate calibration, with the potential sacrifice of performance.

2. Some of the baselines require making decisions on parameter or the choice of kernels. It is not clear how they are chosen in the experiments. This should be made more transparent.

3. Posthoc methods usually can be combined with train-time methods. The evaluation will be more comprehensive if these variants are considered.